# Frequency, active infection and load of *Leishmania infantum* and associated histological alterations in the genital tract of male and female dogs

Viviane Cardoso Boechat[1], Sandro Antonio Pereira[1‡], Artur Augusto Velho Mendes Júnior[1‡], Shanna Araujo dos Santos[1‡], Luciana de Freitas Campos Miranda[2‡], Fabiano Borges Figueiredo[3‡], Luiz Claudio Ferreira[4‡], Francisco das Chagas de Carvalho Rodrigues[4‡], Raquel de Vasconcellos Carvalhaes de Oliveira[5‡], Rayane Teles -de-Freitas[6‡], Rafaela Vieira Bruno[6,7‡], Fernanda Nazaré Morgado[8‡], Rodrigo Caldas Menezes[1]*

1 Laboratório de Pesquisa Clínica em Dermatozoonoses em Animais Domésticos, Instituto Nacional de Infectologia Evandro Chagas, Fundação Oswaldo Cruz, Rio de Janeiro, Brazil, 2 Laboratório de Pesquisa Clínica e Vigilância em Leishmanioses, Instituto Nacional de Infectologia Evandro Chagas, Fundação Oswaldo Cruz, Rio de Janeiro, Brazil, 3 Instituto Carlos Chagas, Fundação Oswaldo Cruz, Curitiba, Paraná, Brazil, 4 Serviço de Anatomia Patológica, Instituto Nacional de Infectologia Evandro Chagas, Fundação Oswaldo Cruz, Rio de Janeiro, Brazil, 5 Laboratório de Epidemiologia Clínica, Instituto Nacional de Infectologia Evandro Chagas, Fundação Oswaldo Cruz, Rio de Janeiro, Brazil, 6 Laboratório de Biologia Molecular de Insetos, Instituto Oswaldo Cruz, Fundação Oswaldo Cruz, Rio de Janeiro, Brazil, 7 Instituto Nacional de Ciência e Tecnologia em Entomologia Molecular (INCT-EM)/CNPq, Rio de Janeiro, Brazil, 8 Laboratório de Pesquisa em Leishmaniose, Instituto Oswaldo Cruz, Fundação Oswaldo Cruz, Rio de Janeiro, Brazil

☯ These authors contributed equally to this work.
‡ These authors also contributed equally to this work.
* rodrigo.menezes@ini.fiocruz.br

**Data Availability Statement:** All relevant data are within the paper.

## Abstract

Visceral leishmaniasis caused by the protozoan *Leishmania infantum* is a zoonosis. The domestic dog is the primary reservoir in urban areas. This study aimed to evaluate the frequency, active infection and load of *L. infantum* in the genital tract of male and female dogs seropositive for this parasite, as well as to identify histological genital alterations associated with this protozoan. We studied 45 male and 25 female *L. infantum*-seropositive noncastrated dogs from the same endemic area in Brazil. Tissue samples from the testis, epididymis, prostate, vulva, vagina, and uterus were examined by singleplex qPCR and parasitological tests (histopathology, immunohistochemistry, and parasitological culture). The latter were performed for the detection of active infection (parasites able to multiply and to induce lesions). Forty-four (98%) males and 25 (100%) females were positive for *L. infantum* in the genital tract (epididymis: 98%; vulva: 92%; vagina: 92%; testis: 91%; uterus: 84%; prostate: 66%). Active infection in the genital tract was confirmed in 69% of males and 64% of females (32% in the uterus). Parasite loads were similar in the testis, vulva, epididymis and vagina and lower in the prostate. Only the parasite load in the vagina was significantly associated with the number of clinical signs. Granulomatous inflammation predominated in all organs, except for the prostate. Only in the testis and epididymis was

**Funding:** This study was supported by the state funding agency Fundação Carlos Chagas Filho de Amparo à Pesquisa do Estado do Rio de Janeiro (Grants: CNE E-26/203.069/2016 and TCT E-26/202.561/2017), http://www.faperj.br, and by Coordenação de Aperfeiçoamento de Pessoal de Nível Superior (CAPES), Brazil (Finance Code 001), https://www.capes.gov.br. FBF and RCM are recipients of productivity fellowships from Conselho Nacional de Desenvolvimento Científico e Tecnológico (CNPq), Brazil, http://www.cnpq.br. The funders had no role in study design, data collection and analysis, decision to publish, or preparation of the manuscript.

**Competing interests:** The authors have declared that no competing interests exist.

the inflammatory infiltrate significantly more intense among dogs with a higher parasite load in these organs. The high frequency, detection of active infection and similarity of *L. infantum* loads in the genital tract of infected males and females suggest the potential of venereal transmission of this parasite by both sexes and of vertical transmission by females in the area studied. Additionally, vertical transmission may be frequent since active *L. infantum* infection was a common observation in the uterus.

## Introduction

Canine visceral leishmaniasis (CVL) caused by the protozoan *Leishmania infantum* is a zoonosis in which the domestic dog (*Canis familiaris*) is the primary reservoir in urban areas [1]. In addition to classical transmission through the bite of infected sandflies, venereal and vertical transmission of the disease is possible [2–5]. In the genital tract of dogs, amastigote forms or DNA of *Leishmania* have been detected in the testis, epididymis, prostate, glans penis, prepuce, scrotum, smegma, semen, vulva, vagina, vaginal secretion, placenta, uterus, uterine tubes, and ovaries [2, 4, 6–16].

An accurate marker of *L. infantum* infectivity in dogs is the parasite load that can be evaluated by singleplex quantitative polymerase chain reaction (qPCR). In addition to providing a diagnostic result, this technique allows to determine the number of copies of the DNA target sequence [17]. In the male genital tract of dogs naturally infected with *L. infantum*, the highest parasite load was detected in the testis and epididymis and was not correlated with inflammatory lesions in these organs [13]. In the female genital tract, the *L. infantum* load was reported to be lower than in the male genital tract and was more intense in the vulva and vagina, but was not associated with inflammatory intensity in these organs [14]. Some authors suggest tropism of *L. infantum* for the male genital tract based on the high frequency of detection of this parasite in this tract and low frequencies in the female genital tract [2, 6, 18]. In addition, venereal transmission of *L. infantum* in dogs has only been demonstrated from male to female [3]. However, other studies found a high frequency of parasitism in the genital tract of females, suggesting tropism of *L. infantum* also for the female genital tract and the possibility of venereal transmission from female to male [11, 14]. The detection of vaginal secretion contaminated with *L. infantum* in naturally infected bitches reinforces the hypothesis of venereal transmission from female to male [16]. In an experimental study, one of eight *L. infantum*-free male BALB/c mice used to breed females infected with this parasite tested positive for *Leishmania* DNA by PCR, suggesting that venereal transmission from female to male may occur in mice [19]. Taken together, these findings demonstrate the lack of consensus regarding the role of females in venereal transmission of *L. infantum* to males, which requires further investigation.

The quantification of *L. infantum* load by qPCR in conjunction with the evaluation of active infection (detection of parasites able to multiply and to induce lesions) in the genital tract by histological techniques and parasitological culture are fundamental to determine the potential of venereal or vertical transmission of the parasite between dogs. However, studies on the frequency of *L. infantum* in the genital tract of dogs that simultaneously evaluate parasite load, the presence of active infection and histological alterations associated with this parasite are scarce [13, 14]. In addition, these studies have evaluated small populations and only males or females from different endemic areas.

Therefore, this study aims to describe the frequency, presence of active infection and parasite load of *L. infantum*, as well as the histological alterations associated with parasitism in different organs of the genital tract, in male and female dogs seropositive for this parasite from the same endemic area.

## Methods

### Animals

This was a descriptive study conducted from August 2015 to December 2016 that used a non-probability sample of 70 dogs (45 males and 25 females). The sample included 57 mongrels, three Miniature Pinschers, three Rottweilers, two Labrador Retrievers, two Cane Corsos, one German Shepherd, one Dachshund, and one Pit Bull. The age of the dogs ranged from one to seven years in 60 (86%) animals, eight to 11 years in seven (10%), and 12 to 14 years in three (4%). The dogs were from the municipality of Barra Mansa (22˚32'25.19" S and 44˚10'35.33" W), state of Rio de Janeiro, Brazil, an area endemic for visceral leishmaniasis with reports of human and canine cases and presence of the vector *Lutzomyia longipalpis* [20, 21]. All dogs were privately owned and tested seropositive for anti-*Leishmania* antibodies by the rapid dual-path platform (TR DPP®) assay [22] and enzyme immunoassay (ELISA) [23], both produced by BioManguinhos (Fiocruz, Rio de Janeiro, Brazil). These two serological tests and diagnostic criteria were used because they are accurate [24] and are recommended by the Brazilian Ministry of Health for considering a dog to be infected with *L. infantum* and for submitting it to euthanasia as a control measure for CVL [25].

### Sample collection

Since they tested positive, the dogs were sent by the Municipal Health Department of Barra Mansa to be euthanized at the Evandro Chagas National Institute of Infectious Diseases (INI--Fiocruz). Euthanasia was performed according to the recommendations of the Brazilian Ministry of Health for the control of visceral leishmaniasis [1], and the owners provided signed consent. The dogs were not housed for any period of time prior to euthanasia.

Immediately after arrival at INI-Fiocruz, the dogs were restrained mechanically and first evaluated clinically, including inspection of the skin and mucosae and palpation of superficial lymph nodes and abdominal organs. The following clinical signs of CVL were considered: thinness or cachexia; diffuse or localized hair loss; cutaneous lesions such as ulcers and desquamation; onychogryphosis; enlargement of the superficial lymph nodes, liver or spleen on palpation; pale ocular or oral mucosae, and skeletal muscle hypotrophy [26, 27]. The animals were divided into two groups according to the clinical signs of CVL: no or few clinical signs (up to three clinical signs) and multiple clinical signs (more than three clinical signs).

After clinical evaluation, the dogs were sedated by intramuscular administration of ketamine hydrochloride (10 mg/kg) and acepromazine maleate (0.2 mg/kg), euthanized with an intravenous overdose of sodium thiopental and potassium chloride, and necropsied. The euthanasia procedure was performed in accordance with the guidelines of the Federal Council on Veterinary Medicine of Brazil [28]. During necropsy, three fragments were collected from each organ of the male (right and left epididymis, right and left testis, and prostate) and female genital tract (vulva, caudal vagina adjacent to the vulva, and uterine corpus). One fragment was immersed in sterile saline and cultured for the detection of *Leishmania*. The second fragment was stored at -20˚C for subsequent *L. infantum* DNA detection by singleplex qPCR. The third fragment was fixed in 10% buffered formalin and embedded in paraffin [29] for immunohistochemistry (IHC) and histopathology (HP). In the case of double organs, one fragment of the right and left testis and epididymis each was collected and examined separately by HP

and IHC. For parasitological culture and qPCR, a pool of samples containing one fragment of the right and left testis and a pool of samples containing one fragment of the right and left epididymis were examined.

## Parasitological culture and identification of *Leishmania* species

The fragments were cultured at 26–28°C in Novy-MacNeal-Nicolle medium plus Schneider's *Drosophila* medium (Sigma-Aldrich®, St. Louis, Missouri, USA) supplemented with 10% fetal bovine serum and penicillin and streptomycin as antibiotics [30]. The parasitological culture was considered positive if promastigote forms of *Leishmania* grew in the culture medium up to 30 days and were visualized by conventional optical microscopy [30]. The detailed protocol of parasite isolation in culture is registered at https://dx.doi.org/10.17504/protocols.io.22tggen. Parasites isolated in culture were identified as *L. infantum* by multilocus enzyme electrophoresis (MLEE) [31].

## Singleplex qPCR for the diagnosis and quantification of *Leishmania infantum* load

First, DNA was extracted from the organs using the DNeasy® Blood & Tissue Kit (Qiagen, Hilden, Germany) and QIAcube semi-automated extraction platform (Qiagen) according to manufacturer instructions. Tissue fragments $\leq$ 25 mg were used. The DNA was quantified in a Qubit® 2.0 fluorometer (Thermo Fisher Scientific, Waltham, Massachusetts, USA) using the dsDNA HS Qubit® Kit (Thermo Fisher Scientific) according to manufacturer instructions. Amplification was performed with the StepOne™ system (Applied Biosystems, Carlsbad, CA, USA) using one primer pair and the TaqMan® MGB probe (Thermo Fisher Scientific), following a previously described protocol [32].

Parasite load was quantified by constructing a standard curve with serial dilutions ($10^1$ to $10^5$ parasites) of *L. infantum* DNA (MHOM/BR/1974/PP75). Positive and negative controls were included in each amplification plate and a threshold of 0.1 was established. The DNA of 1 x $10^5$ promastigote forms of *L. infantum* obtained by parasitological culture was used as positive control and ultrapure water as negative control. Samples in which DNA amplification occurred after the 37th cycle were classified as undetectable. The *L. infantum* load was expressed as the natural logarithm of the number of parasite genome equivalents (gEq)/ng [32].

Samples with an undetectable result in the amplification were submitted to DNA quality testing in another qPCR using the TaqMan® Gene Expression Assay (Applied Biosystems) [32]. The results are expressed as positive or negative and samples showing amplification were considered free of DNA degradation and PCR inhibitors.

For the comparison of parasite load between the different organs of the male and female genital tract, only dogs positive for *L. infantum* DNA in the genital tract by qPCR were analyzed. In turn, for the comparison of parasite load between the different organs of the male and female genital tract according to the clinical signs of the dogs and intensity of inflammatory infiltration in these organs, only dogs that tested positive for *L. infantum* by qPCR or parasitological methods were analyzed.

## Immunohistochemistry

Histological sections (5 μm thick) were mounted on silanized microscope slides and submitted to deparaffinization, rehydration, blocking of endogenous peroxidase, antigen retrieval, blockade of nonspecific protein binding, and incubation with polyclonal rabbit anti-*Leishmania* serum diluted 1:500 following a previously described protocol [11]. A biotin-streptavidin-

peroxidase system (UltraVision™ Plus Large Volume Detection System Anti-Polyvalent, HRP, Ready-to-Use Kit, Thermo Fisher Scientific) was used for the detection of *Leishmania* according to manufacturer recommendations. Histological sections of organs intensely parasitized with amastigote forms of *Leishmania* were incubated with non-immune homologous serum as negative control and with polyclonal rabbit anti-*Leishmania* serum as positive control. The testis and epididymis were considered positive for *Leishmania* if amastigote forms of this parasite were detected in at least one fragment of the right or left organ. In addition, the frequencies of positivity for amastigote forms of *L. infantum* in the right and left testis and epididymis were evaluated.

## Histopathology

Histological sections (5 μm thick) were mounted on microscope slides and stained with hematoxylin-eosin [29]. The inflammatory infiltrate in the organs was classified as follows: granulomatous, predominance of cells of the monocyte-macrophage system (activated macrophages, epithelioid macrophages, or multinucleate giant cells); non-granulomatous, predominance of other types of inflammatory cells (lymphocytes, plasma cells, and neutrophils). The intensity of inflammatory infiltration was classified as absent or mild (absent cellular infiltrate or mild and dispersed foci) and moderate to intense (dense and diffuse cellular infiltrate). The frequency of histological alterations in the genital tract was compared between the group of dogs that tested positive and the group of dogs that tested negative for *Leishmania* by the parasitological methods (parasitological culture, HP or IHC) in these organs. This comparison was done to verify the association between lesions with active *Leishmania* infection.

The testis and epididymis were considered positive for *Leishmania* if amastigote forms of this parasite were detected in at least one fragment of the right or left organ. For calculation of the frequency of each histological alteration in the testis and epididymis, the histological alteration was considered present if detected in at least one fragment of the right or left organ. The intensity of inflammatory infiltration was classified in the organ (right or left) with the most intense inflammatory infiltrate. In addition, the frequencies of positivity for amastigote forms of *L. infantum* and of inflammatory infiltrates associated with amastigote forms of this parasite in the right and left testis and epididymis were evaluated.

## Statistical analysis

Data were analyzed using the free R software, version 3.5.1 [33]. The following variables were reported as simple frequencies: clinical classification (no or few clinical signs of CVL; multiple clinical signs of CVL), positive results with the diagnostic techniques, and histological alterations. Boxplots were used for comparison of log parasite loads in the samples of each organ. The Shapiro-Wilk test rejected the normality assumption of the parasite load data. Thus, the correlation between *L. infantum* loads in the genital tract samples was evaluated using the nonparametric Spearman correlation coefficient. The correlations ranged from -1 to 1, with positive values indicating a positive correlation and negative values an inverse correlation. In addition, the nonparametric Mann-Whitney test was applied to compare log parasite loads among all organs of the genital tract (male and female ones) and according to the intensity of clinical signs and inflammatory infiltrate. The level of agreement between the results of the diagnostic techniques (positive/negative) for detection of *Leishmania* infection in all organs of the genital tract was assessed by calculating Cohen's kappa (k) statistic, considering the classification proposed by Landis and Koch [34]. A level of significance of 5% was adopted in this study.

### Ethics statement

This study was carried out in strict accordance with the recommendations of the Brazilian Ministry of Health and the Federal Council on Veterinary Medicine, with permission of the owners. The study protocol was approved by the Ethics Committee on Animal Use of the Oswaldo Cruz Foundation (CEUA/Fiocruz; Permit Number: LW-54/13).

## Results

Clinical examination of the 70 dogs revealed clinical signs in 59 (84.3%), 38 (54.3%) males and 21 (30.0%) females. The following clinical signs were observed: thinness (26 males and 10 females), skin ulcer (25 males and 9 females), hair loss (23 males and 10 females), onychogryphosis (20 males and 11 females), lymphadenomegaly (21 males and 10 females), furfuraceous desquamation (21 males and 5 females), splenomegaly (11 males and 7 females), cachexia (8 males and 4 females), keratoconjunctivitis (8 males and 3 females), and hepatomegaly (2 males and 2 females). Thirty-four (57.6%) of the 59 dogs exhibited up to three clinical signs and 25 (42.4%) had more than three clinical signs compatible with CVL.

In the genital tract of the 45 males and 25 females, 98% of males and 100% of females tested positive for *L. infantum* by at least one of the diagnostic techniques used (Table 1). The only negative animal in all assays was a male without clinical signs.

Table 2 shows the level of agreement between the results of the diagnostic techniques for detection of *Leishmania* infection in all organs of the genital tract using Cohen's kappa statistic.

In the testis of 17 dogs positive for amastigote forms of *Leishmania* by HP or IHC, bilateral infection was observed in 11 (65%) dogs and unilateral in four (23%). In two dogs (12%), only one testis was examined because one dog was monorchid and the other a unilateral cryptorchid. In the epididymis of 21 dogs positive for amastigote forms of *Leishmania* by HP or IHC, bilateral infection was observed in 14 (67%) dogs and unilateral in five (24%). In the same two dogs (9%) that were monorchid or cryptorchid, only one epididymis was examined.

The combined parasitological techniques (parasitological culture, HP, and IHC) detected active *Leishmania* infection in the genital tract of 69% of males (*n* = 31) and 64% of females (*n* = 16). The frequencies of a positive parasitological result per organ were: 64% (*n* = 16) in

**Table 1. Frequency of a positive *Leishmania* result in genital tract organs of male and female dogs seropositive for *L. infantum* detected by different diagnostic techniques.**

| Organs | Frequency of positivity | | | | | | | | | |
|---|---|---|---|---|---|---|---|---|---|---|
| | qPCR[a] | | IHC[b] | | HP[b] | | PC[a] | | Total | |
| | n | % | n | % | n | % | n | % | n | % |
| Testis (N = 45) | 41 | 91 | 17 | 38 | 12 | 27 | 23 | 51 | 41 | 91 |
| Epididymis (N = 45) | 44 | 98 | 21 | 47 | 16 | 35 | 12 | 27 | 44 | 98 |
| Prostate (N = 45) | 29 | 64 | 2 | 4 | 0 | 0 | 11[c] | 25[c] | 29 | 64 |
| Vulva (N = 25) | 22 | 88 | 9 | 36 | 9 | 36 | 12 | 48 | 23 | 92 |
| Vagina (N = 25) | 21 | 84 | 6 | 24 | 4 | 16 | 12 | 48 | 23 | 92 |
| Uterus (N = 25) | 21 | 84 | 3 | 12 | 3 | 12 | 7 | 28 | 21 | 84 |

qPCR: quantitative PCR; HP: histopathology, IHC: immunohistochemistry; PC: parasitological culture; N: number of samples of each genital tract organ examined; n: number of genital tract samples with positive *Leishmania* result; %: percentage; Total: positive result with at least one of the diagnostic techniques.

[a]A pool of samples containing one fragment of the right and left testis and a pool of samples containing one fragment of the right and left epididymis were examined.

[b]The testis and epididymis were considered positive for *Leishmania* if amastigote forms of this parasite were detected in at least one fragment of the right or left organ.

[c]Forty-four prostate samples were examined by PC due to the loss of one sample.

**Table 2. Agreement between the different diagnostic techniques for detection of *Leishmania* infection in genital tract organs of male and female dogs.**

| Techniques compared | Kappa index/organ (agreement [b]) | | | | | |
|---|---|---|---|---|---|---|
| | Testis | Epididymis | Prostate | Vulva | Vagina | Uterus |
| PC vs IHC | 0.38 (moderate) | 0.40 (fair) | 0.08 (slight) | 0.77 (substantial) | 0.55 (moderate) | 0.86 (almost perfect) |
| PC vs HP | 0.52 (moderate) | 0.38 (fair) | _[a] | 0.84 (almost perfect) | 0.55 (moderate) | 0.86 (almost perfect) |
| PC vs qPCR | 0.18 (slight) | 0.02 (slight) | 0.22 (fair) | 0.68 (substantial) | 0.65 (substantial) | 0.58 (moderate) |
| IHC vs HP | 0.55 (moderate) | 0.77 (substantial) | _[a] | 0.87 (almost perfect) | 0.77 (substantial) | 1.00 (almost perfect) |
| qPCR vs IHC | 0.11 (slight) | 0.04 (slight) | 0.05 (slight) | 0.54 (moderate) | 0.49 (moderate) | 0.48 (moderate) |
| qPCR vs HP | 0.07 (slight) | 0.02 (slight) | _[a] | 0.60 (moderate) | 0.50 (moderate) | 0.48 (moderate) |

qPCR: quantitative PCR; HP: histopathology; IHC: immunohistochemistry; PC: parasitological culture; vs: versus.

[a]The kappa index could not be calculated because all prostate samples were negative for amastigote forms of *Leishmania* by histopathology.

[b]Level of agreement according to Landis and Koch [34].

the vagina, 58% ($n = 26$) in the testis, 56% ($n = 14$) in the vulva, 51% ($n = 23$) in the epididymis, 32% ($n = 8$) in the uterus, and 27% ($n = 12$) in the prostate. In the 23 dogs (14 males and nine females) in which the parasitological tests did not detect *Leishmania* in the genital tract, *L. infantum* DNA was found in at least one of the genital samples.

The qPCR technique detected *L. infantum* DNA in at least one genital tract sample in 98% of males ($n = 44$) and 96% of females ($n = 24$). In the female negative by qPCR, vulva and vagina tested positive for amastigote forms of *Leishmania* by IHC.

Fig 1 illustrates the *L. infantum* load in the male and female genital tract. A positive correlation was observed between parasite load in the testis and epididymis ($r = 0.80$, $P < 0.001$) and in the vulva and vagina ($r = 0.53$, $P = 0.01$). There was insufficient statistical evidence to assert a difference in parasite load between the testis and epididymis ($P = 0.665$), testis and vulva ($P = 0.746$), testis and vagina ($P = 0.292$), epididymis and vulva ($P = 0.437$), epididymis and vagina ($P = 0.484$), vagina and vulva ($P = 0.162$), and vagina and uterus ($P = 0.241$).

The *L. infantum* load in the uterus was significantly lower than that detected in the testis ($P = 0.037$) and vulva ($P = 0.008$). The parasite load in the prostate was significantly lower than that detected in the testis ($P < 0.001$), epididymis ($P < 0.001$), uterus ($P = 0.037$), vulva ($P < 0.001$), and vagina ($P = 0.002$).

The comparison of *L. infantum* load in the male and female genital tracts according to clinical classification is described in Table 3. Only parasite load in the vagina was significantly associated with the number of clinical signs.

In the genital tract of the 47 dogs that tested positive for *Leishmania* by the parasitological methods, histological alterations were observed in at least one of the organs examined in all 31 males (Fig 2) and 16 females (Fig 3). Among the 23 dogs with negative parasitological tests, 10 males and 9 females exhibited histological alterations in at least one of the organs examined. In these 19 dogs, the inflammatory infiltrate observed in the genital tract organs was associated with the detection of *L. infantum* DNA in these organs. Table 4 shows the frequencies of histological alterations in the genital tract of dogs that tested positive and negative for *Leishmania* by the parasitological methods (parasitological culture, HP or IHC) in these organs.

A bilateral inflammatory infiltrate associated with amastigote forms of *Leishmania* detected by HP or IHC was observed in the testis of 11 dogs and in the epididymis of 14 dogs and a unilateral infiltrate in the testis of four dogs and in the epididymis of five dogs.

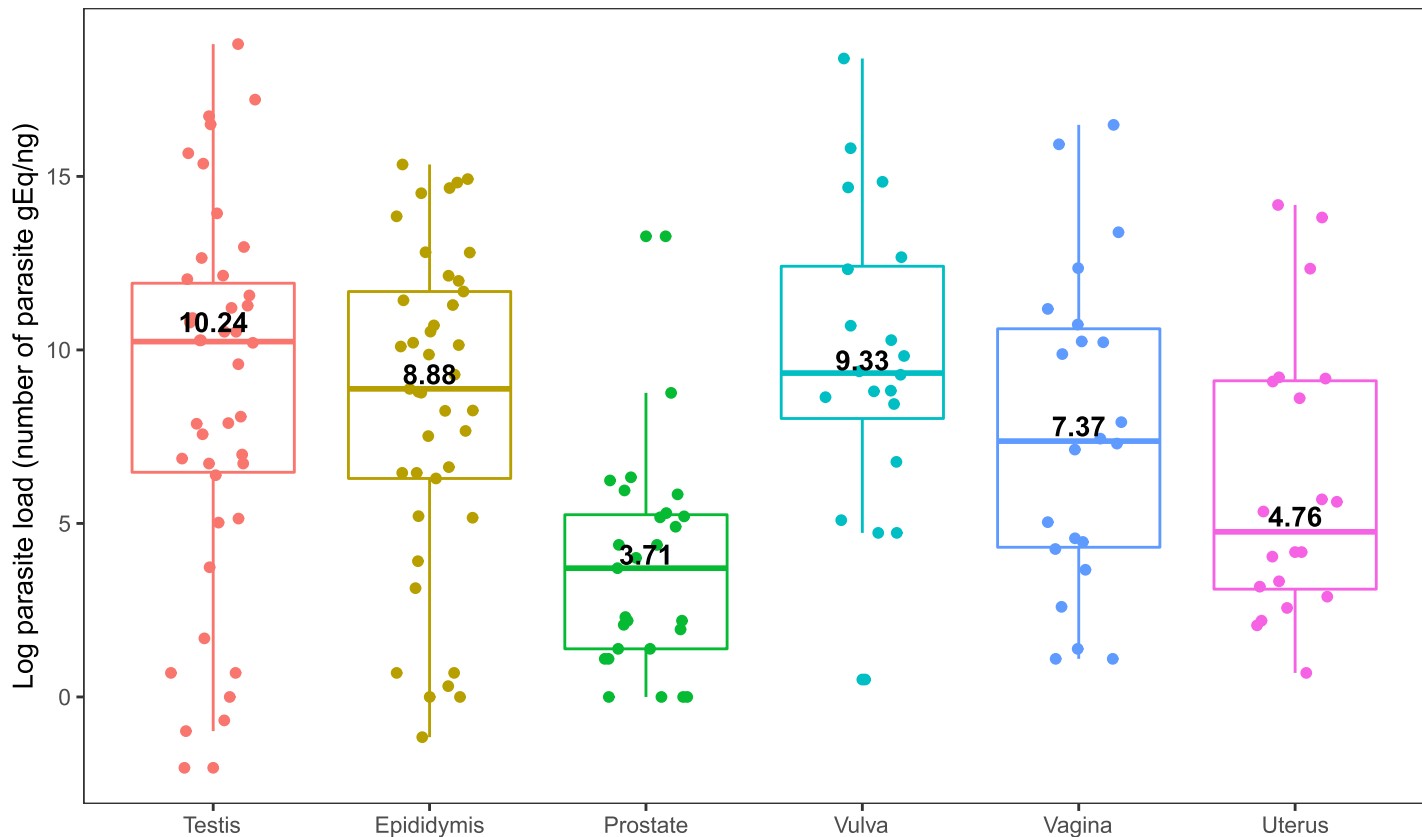

**Fig 1.** *Leishmania infantum* **load expressed as the natural logarithm of the number of genome equivalents/nanogram DNA (gEq/ng) in testis (a pool of right and left organs), epididymis (a pool of right and left organs), prostate, vulva, vagina, and uterus of 68 dogs positive for *L. infantum* DNA in the genital tract by qPCR.** The colored horizontal lines indicate the median parasite load. The vertical lines indicate the interquartile range. Colored dots represent the parasite load of each dog.

**Table 3.** *Leishmania infantum* load in the genital tract of naturally infected dogs according to clinical classification.

| Organ | *L. infantum* load[a] | | | | |
| --- | --- | --- | --- | --- | --- |
| | Dogs with no or few clinical signs[b] (N = 44)[d] | | Dogs with multiple clinical signs[c] (N = 25) | | *P* value |
| | Median | Range | Median | Range | |
| Testis | 9.89 | -0.68–16.74 | 10.52 | -2.04–18.81 | 0.245 |
| Epididymis | 9.04 | -1.16–15.34 | 10.10 | 0.31–14.92 | 0.388 |
| Prostate | 4.38 | 1.39–8.76 | 4.01 | 1.10–13.27 | 0.504 |
| Vulva | 8.82 | 0.5–18.4 | 11.07 | 4.73–14.84 | 0.343 |
| Vagina | 6.08 | 1.1–15.92 | 10.96 | 4.26–16.49 | 0.032 |
| Uterus | 4.75 | 0.69–13.82 | 6.57 | 3.18–14.18 | 0.364 |

N: number of dogs.

[a]*L. infantum* load expressed as the natural logarithm of the number of genome equivalents/nanogram DNA.

[b]Up to three clinical signs compatible with canine visceral leishmaniasis.

[c]More than three clinical signs compatible with canine visceral leishmaniasis.

[d]One male dog was excluded because it tested negative for *L. infantum* by qPCR and parasitological methods.

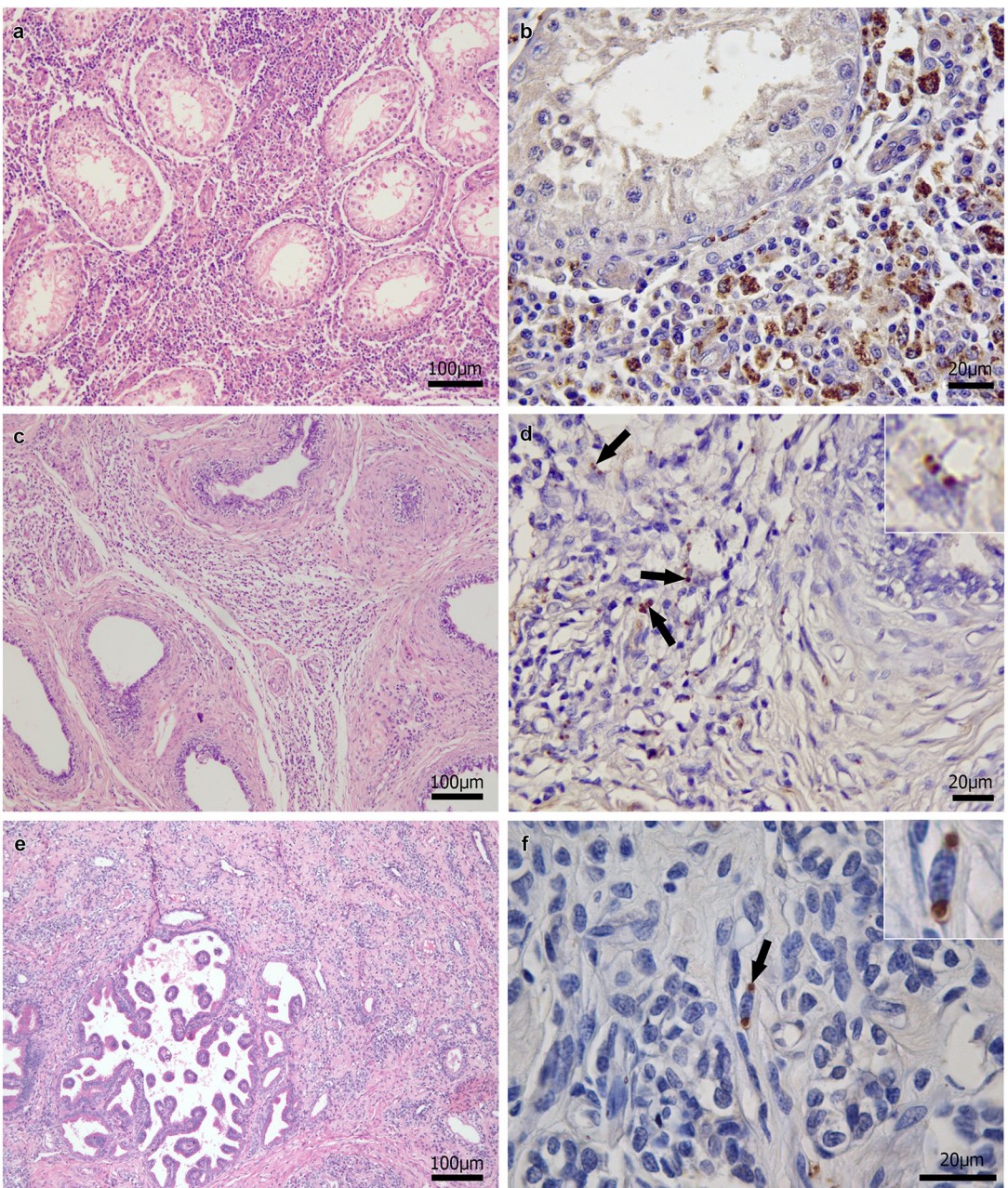

**Fig 2. Histological findings in the genital tract of male dogs naturally infected with *Leishmania infantum*.** (a) Marked diffuse granulomatous interstitial orchitis. (b) Many brown-stained *Leishmania* amastigotes in the cytoplasm of macrophages in testicular interstitial tissue. (c) Marked diffuse granulomatous interstitial epididymitis. (d) Brown-stained *Leishmania* amastigotes in the cytoplasm of macrophages (arrows and inset) in the epididymis. (e) Prostate showing marked diffuse granulomatous inflammation, fibrosis, and glandular atrophy. (f) Brown-stained *Leishmania* amastigotes in the cytoplasm of macrophages (arrow and inset) in the prostate. Hematoxylin-eosin staining (a,c,e); immunohistochemistry (b,d,f).

Table 5 describes the intensity of the inflammatory infiltrate according to *L. infantum* load in the genital tract organs of the dogs studied. Only in the testis and epididymis was the intensity of the inflammatory infiltrate significantly associated with the parasite load in these tissues.

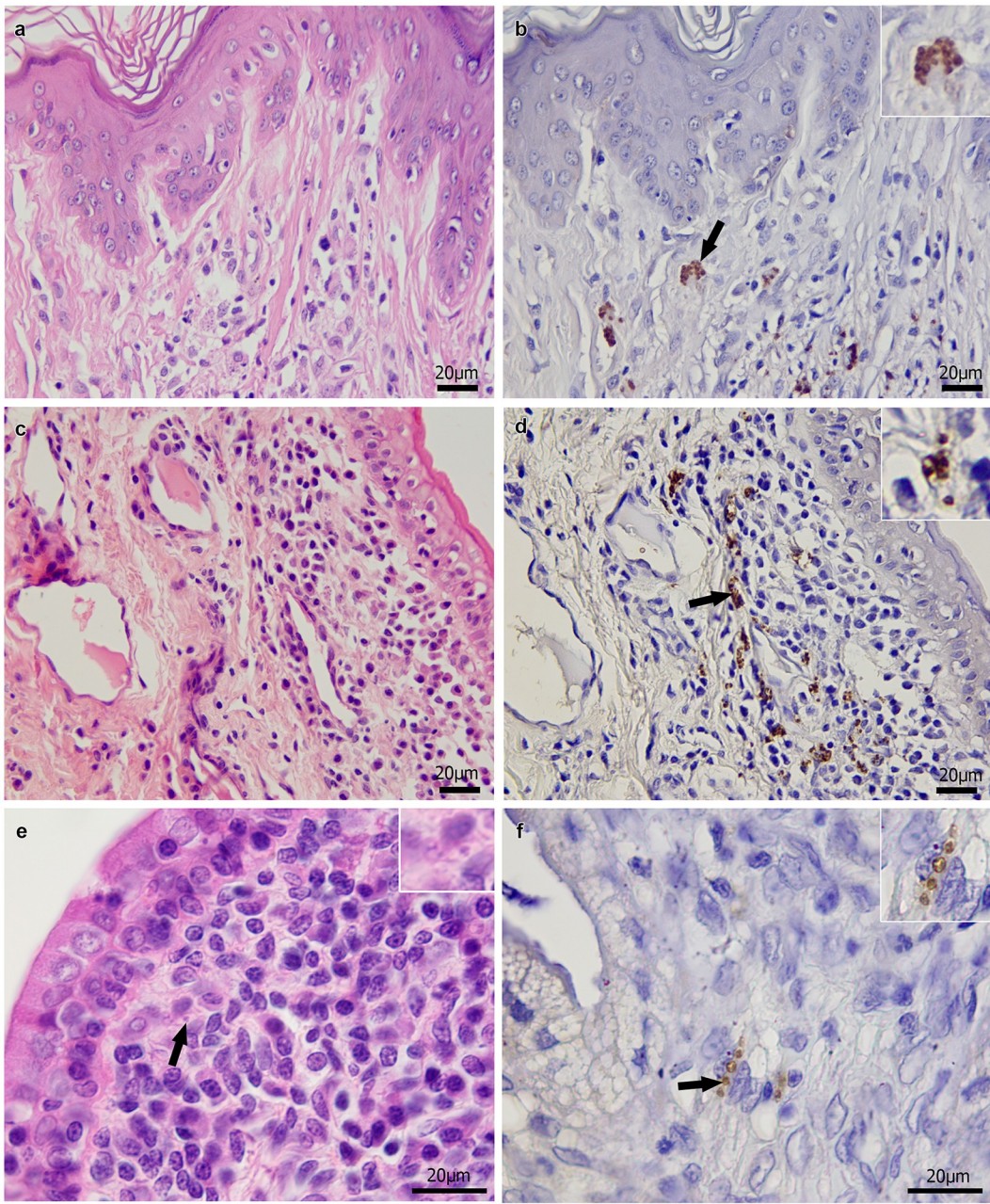

**Fig 3. Histological findings in the genital tract of female dogs naturally infected with *Leishmania infantum*.** (a) Moderate diffuse granulomatous vulvitis in the dermis and acanthosis and hyperkeratosis in the epidermis. (b) Brown-stained *Leishmania* amastigotes in the cytoplasm of macrophages (arrow and inset) in the vulva. (c) Marked diffuse granulomatous vaginitis in the lamina propria. (d) Several brown-stained *Leishmania* amastigotes in the cytoplasm of macrophages (arrow and inset) in the vagina. (e) Marked diffuse granulomatous endometritis in the lamina propria with an amastigote form of *Leishmania* (arrow and inset) in the cytoplasm of macrophages. (f) Brown-stained *Leishmania* amastigotes in the cytoplasm of macrophages (arrow and inset) in the uterus. Hematoxylin-eosin staining (a,c,e); immunohistochemistry (b,d,f).

## Discussion

The findings of the present study suggest that *L. infantum* parasitism in organs of the genital tract of dogs is not influenced by sex since a high frequency of the parasite was observed in

**Table 4. Histological alterations in each genital tract organ of dogs seropositive for *Leishmania infantum* according to a positive or negative result by the parasitological methods.**

| Organ | Histological alterations | Parasitological method for *Leishmania*[a] | | | |
|---|---|---|---|---|---|
| | | Positive | | Negative | |
| | | n/total | % | n/total | % |
| Testis (N = 45)[b] | Granulomatous inflammation | 21/26 | 80.7 | 5/19 | 26.3 |
| | Seminiferous tubule degeneration | 18/26 | 69.2 | 3/19 | 15.7 |
| | Seminiferous tubule atrophy | 14/26 | 53.8 | 4/19 | 21.0 |
| | Absence of spermatogenesis | 10/26 | 38.4 | 0/19 | 0.0 |
| | Fibrosis | 4/26 | 15.3 | 1/19 | 5.2 |
| | Non-granulomatous inflammation | 2/26 | 7.7 | 0/19 | 0.0 |
| | Necrosis | 1/26 | 3.8 | 0/19 | 0.0 |
| Epididymis (N = 45)[b] | Granulomatous inflammation | 21/23 | 91.3 | 11/22 | 50.0 |
| | Non-granulomatous inflammation | 2/23 | 8.6 | 2/22 | 9.0 |
| Prostate (N = 45) | Non-granulomatous inflammation | 8/12 | 66.6 | 10/33 | 30.3 |
| | Fibrosis and glandular atrophy | 6/12 | 50.0 | 2/33 | 6.0 |
| | Granulomatous inflammation | 1/12 | 8.3 | 1/33 | 3.0 |
| Vulva (N = 25) | Granulomatous inflammation | 9/14 | 64.2 | 7/11 | 63.6 |
| | Non-granulomatous inflammation | 5/14 | 35.7 | 1/11 | 9.0 |
| | Hyperkeratosis | 2/14 | 14.2 | 0/11 | 0.0 |
| Vagina (N = 25) | Granulomatous inflammation | 4/16 | 25.0 | 1/9 | 11.1 |
| | Non-granulomatous inflammation | 4/16 | 25.0 | 0/9 | 0.0 |
| Uterus (N = 25) | Granulomatous inflammation | 3/8 | 37.5 | 0/17 | 0.0 |
| | Non-granulomatous inflammation | 1/8 | 12.5 | 0/17 | 0.0 |

N: number of samples of each genital tract organ examined; n: number of genital tract samples with histological alterations; %: percentage.

[a]Parasitological method: histopathology, immunohistochemistry, and parasitological culture.

[b]The histological alteration was considered present if detected in at least one fragment of the right or left organ.

both male and female organs. Previous studies involving dogs with parasitological confirmation of *L. infantum* have also reported a high frequency of positive results in the genital tract of males (94.7%) and females (90.0%) by IHC and qPCR [13, 14]. In another study involving dogs with parasitological confirmation that originated from the same area as the animals of the

**Table 5. *Leishmania infantum* load according to the intensity of the inflammatory infiltrate in genital tract samples of naturally infected dogs.**

| Sex | Organ | *Leishmania infantum* load[a] | | | | | | |
|---|---|---|---|---|---|---|---|---|
| | | Absent or mild InI | | | Moderate to intense InI | | | |
| | | n | Med | Range | n | Med | Range | *P* value |
| Male (N = 44[b]) | Testis[c] | 24 | 6.98 | -2.04–13.93 | 21 | 10.86 | -0.98–18.81 | 0.017 |
| | Epididymis[c] | 16 | 3.13 | -1.15–12.80 | 29 | 10.33 | 3.91–15.34 | < 0.001 |
| | Prostate | 33 | 3.00 | 0.0–13.27 | 12 | 4.00 | 0.0–8.76 | 0.857 |
| Female (N = 25) | Vulva | 7 | 10.28 | 4.72–18.39 | 18 | 8.82 | 0.50–15.81 | 0.275 |
| | Vagina | 20 | 7.44 | 1.09–15.92 | 5 | 7.30 | 2.59–16.48 | 0.938 |
| | Uterus | 23 | 4.17 | 0.69–14.17 | 2 | 9.75 | 5.69–13.81 | 0.186 |

N: number of dogs; n: number of samples of each genital tract organ; InI: inflammatory infiltration; Med: median *L. infantum* load.

[a]*L. infantum* load expressed as the natural logarithm of the number of genome equivalents/nanogram DNA.

[b]One male dog was excluded because it tested negative for *L. infantum* by qPCR and parasitological methods.

[c]The intensity of inflammatory infiltration was classified in the organ (right or left) with the most intense inflammatory infiltrate.

present study, the frequencies of positive results in the genital tract were 90% among males and 80% among females [11]. This lower frequency compared to the present study might be due to the exclusive use of parasitological techniques, which are less sensitive than PCR [32]. However, other authors investigating seropositive females suggest that *L. infantum* does not exhibit tropism for the female genital tract since amastigote forms of this parasite were detected only in the vulva, with a low frequency (20%) by IHC and HP, and no specific inflammatory alterations were observed in the genital organs [6]. In contrast, in the present study amastigote forms of *Leishmania* were observed in the uterus and vagina; there was a higher frequency in the vulva (36%), which was associated with a specific inflammatory reaction. The different results reported by Silva et al. [6] might be related to the small number of females studied (n = 10), in addition to the exclusive use of a positive serological indirect fluorescence antibody test as a criterion for inclusion of the animals. The accuracy of this test is lower than that of the combination of TR DPP$^®$ and ELISA [24, 35], which were used in the present study as inclusion criteria. In the present study, the detection of *L. infantum* parasitism in almost 100% of the genital tract of males and females seropositive for this parasite confirms the high specificity already described for the combination of TR DPP$^®$ and ELISA tests [24].

The highest frequency of *L. infantum*-positive specimens was observed for the epididymis, followed by the vagina, vulva, testis, uterus and prostate, considering all diagnostic techniques used in this study. There was a small difference in positive results (1 to 7%) between the testis, epididymis, vagina, and vulva. The prostate exhibited a much lower frequency of positive specimens, with the difference ranging from 18% to 32% compared to the other organs. Other authors [13] also reported similar frequencies of *L. infantum*-positive specimens in the testis (89.5%) and epididymis (84.2%) of dogs. These authors [13] observed a lower frequency of positive results in the prostate (78.9%), but higher than that found in the present study. In bitches, studies found a higher frequency of positive results in the uterus (80%), followed by the vagina (70%) and vulva (60%) [14]. These results differ from the present study in which the frequencies of positive results were much higher in the vulva and vagina compared to the uterus. The small number of males (n = 19) and females (n = 10) examined in these two previous studies [13, 14] may explain these differences in the frequency of positive specimens. In the present study, the level of agreement between the parasitological techniques was better than the level of agreement between qPCR and parasitological techniques in all organs examined, except for the prostate. These results can be explained by the differences in the frequencies of positivity for *Leishmania*. These differences were smaller between parasitological techniques and greater when qPCR and the parasitological techniques were compared. The latter are considered less sensitive than qPCR [32]. Among the genital tract organs, the lowest level of agreement between the diagnostic techniques that could be compared was observed for the prostate. This finding is related to the low frequency of positivity for *Leishmania* in this organ by parasitological techniques.

The frequency of active *L. infantum* infection was similar and high in the male and female genital tract of the animals studied here, as demonstrated by the high percentages of positive results obtained with the parasitological methods. The vagina, testis, vulva and epididymis were the organs showing the highest frequencies of active infection with this parasite, which exceeded 50%. These results corroborate the findings of another study on dogs from the same endemic region that showed higher frequencies of active *L. infantum* infection in the vulva (70%), epididymis (55%), testis (45%), and vagina (45%) when compared to the prostate (15%) and uterus (15%) [11]. Although the uterus was the least parasitized female genital organ in the present study, about 1/3 of bitches had active *L. infantum* infection in this organ. This finding suggests a high risk of vertical transmission of the parasite, a fact already confirmed in dogs [36–41]. Vertical transmission of CVL is the main route of transmission in areas without

demonstration of vector transmission, such as the United States, and may contribute to the maintenance of visceral leishmaniasis in endemic areas [12, 40, 41]. The lower frequency of active *L. infantum* infection in the prostate and the fact that the blood-prostate barrier seems to restrict the passage of leukocytes from the interstitium to the ductal fluid during prostatitis [42] suggest a minor role of this organ in the contamination of dog semen with *L. infantum* when compared to the testis and epididymis. Considering the high frequency of active *L. infantum* infection in the genital tract of male and female dogs in this study, the castration of dogs would be an important control measure of CVL in endemic areas, preventing venereal and vertical transmission.

The four organs with the highest *L. infantum* load, i.e., testis, epididymis, vulva and vagina, also exhibited the highest frequencies of positive results but no significant difference was observed between these organs. This is the first report of similar *L. infantum* loads in organs of the female and male genital tracts of dogs from the same endemic region. The lowest parasite load was found in the prostate and uterus, in agreement with the finding of the lowest frequency of positive results. Other authors also detected a higher parasite load by qPCR in testis and epididymis, although there was no statistically significant difference compared to parasite load in the prostate [13]. In contrast to our study, in females, the *L. infantum* load detected by qPCR in the vulva and vagina was similar to that of the uterus [14]. Studies evaluating parasite loads by IHC also detected higher loads in the testis, epididymis, vulva and vagina [2, 11]. The positive correlation between parasite load in the testis and epididymis and in the vulva and caudal vagina adjacent to the vulva observed in the present study can be explained by the anatomic proximity and communication between these organs [43, 44]. The lower frequency and load of *L. infantum* in the prostate compared to the epididymis and testis observed in this study and in others [4, 11, 15] may be due to a more efficient innate and adaptive immunity against *L. infantum* in this organ. An important mechanism of innate immunity that protects the prostate from infections is the blood-prostate barrier [42, 45]. A temperature lower than the body temperature and reduced immune responsiveness in the testis, as well as the lack of a developed mucosal immune system in the epididymis [45, 46], may predispose these organs to the infection with *L. infantum*.

The venereal transmission of *L. infantum* in dogs has only been demonstrated from male to female [3]. According to some authors [3, 6, 18], venereal transmission of *L. infantum* in dogs tends to be unidirectional, from male to female, due to the exclusive tropism of *L. infantum* for the male genital tract. However, the present findings of a similar parasite load and a high frequency of active *L. infantum* infection in the male and female genital tracts reinforce the hypothesis of venereal transmission also from female to male. This type of transmission, which needs to be confirmed experimentally, has also been suggested in other studies based on the high frequencies of *L. infantum* in the genital tract of females [11, 14]. Furthermore, the isolation of *L. infantum* from vaginal secretion of female dogs and the infection of an *L. infantum*-free male mouse used experimentally to breed females infected with this parasite support the possibility of venereal transmission from female to male [16, 19].

In venereal transmission, infection would first occur in the male and female external genital organs during copulation. In males, the infection would then spread via the hematogenous or lymphatic route from the penis, which shows a high frequency of this parasite in the prepuce, glans penis and smegma [10, 11, 13, 15], to the testis, epididymis and prostate. In females, the infection would spread via the hematogenous or lymphatic route from the vulva and vagina to the uterus, uterine tubes and ovaries. This hypothesis may explain the lower frequency and parasite load in the uterus observed in the present study and in the uterus, uterine tubes and ovaries in the study of Boechat et al. [11] when compared to the external genital organs. According to a previously reported hypothesis, venereal transmission of *L. infantum* from

male to female dogs would occur by transfer of amastigote forms present in the glans penis, prepuce and smegma through contact with the vulvo/vaginal mucosae or with the oral mucosa of females by licking and sniffing of the preputial/penile region, as well as by ejaculation of infected semen into the female genital tract [2, 3, 10, 11]. On the other hand, Magro et al. [16] suggested that venereal transmission of this parasite from female to male would be a result of the sexual behavior of males that sniff and lick the vulva and thus come in contact with positive vaginal secretions, as well as the contact of the penis with these secretions during copulation. In addition to venereal transmission, the exposure of the skin of male and female external genitalia to the bites of infected sandflies can cause infection with *L. infantum*. After infection, the parasite can spread through the blood or lymphatic system to other organs of the genital tract. The high frequencies of positive results in vulvar skin observed in this study and in vulvar skin and prepuce reported in other studies [2, 11, 13, 15] support that this route of infection of the genital tract may occur. However, future experimental studies on the kinetics of genital tract infection with *L. infantum* in dogs are necessary to confirm all of the hypotheses raised in the present study.

In the present study, the parasite load was significantly higher in the vagina of females with multiple clinical signs of CVL compared to those without or with few clinical signs. Although no association of clinical signs with parasite load was observed in the other organs of the genital tract, all except for the prostate exhibited a higher *L. infantum* load in dogs with multiple clinical signs of CVL. Other authors found a significantly higher *L. infantum* load in the testis, epididymis, glans and prepuce of dogs with clinical signs of CVL compared to those without signs of the disease [15]. Thus, the results of that study [15] and the present findings suggest that, among both males and females, the potential of venereal transmission is higher in animals with clinical signs compatible with CVL. However, this hypothesis needs to be confirmed.

The frequency of an inflammatory reaction was much higher in organs that tested positive by the parasitological methods compared to those in which no parasites were detected by these tests. Additionally, granulomatous inflammation, which is expected as a response to the infection of different organs with *L. infantum* in dogs [47], predominated in all organs, except for the prostate. These results indicate that the histological alterations were associated with active *L. infantum* infection. Even in the 19 dogs in which histological alterations were not associated with the detection of *L. infantum* by parasitological methods, the inflammatory reaction was probably associated with this parasite, as *L. infantum* DNA was detected in the affected genital organs of these dogs. In addition, in these 19 dogs, the inflammation may have been caused indirectly by *L. infantum* via immunocomplex deposition [47]. However, the bacterium *Brucella canis* may also cause granulomatous inflammation in the genital tract of dogs [48]. Although unlikely, its participation in this type of inflammation in the present study should therefore not be completely ruled out. The predominance of a non-granulomatous inflammatory reaction in the prostate, as also observed in other studies involving dogs with CVL [9, 13], might be explained by the lower parasite load and lower frequency of *L. infantum*-positive specimens compared to the other organs examined. Other causes of non-granulomatous prostatitis may have also influenced the predominance of this type of inflammatory reaction, such as bacterial infections with *Escherichia coli* or urinary reflux [49]. In the testis, inflammation associated with *L. infantum* infection was the likely cause of degeneration and atrophy of the seminiferous tubules, fibrosis, necrosis and absence of spermatogenesis, in agreement with other studies [2, 7, 11]. These testicular alterations, together with epididymitis, which were mostly bilateral, as well as prostatitis accompanied by fibrosis and glandular atrophy associated with *L. infantum* infection in this study, can compromise semen quality and can cause infertility of dogs with CVL [9, 50, 51]. According to Assis et al. [50], poor semen quality of *L. chagasi* (syn. *L. infantum*)-infected dogs, which is characterized by poor motility, principal piece

defects and detached heads, indicates epididymal dysfunction. Except for the vulva and vagina, inflammation was more intense in genital tract organs with a higher parasite load, although a significant association was only found for the testis and epididymis. In the vulva and vagina, the result may have been influenced by the small number of animals in the groups with an absent or mild inflammatory reaction in the vulva and moderate to intense reaction in the vagina. These results suggest that the intensity of inflammation in the genital tract of dogs is associated with *L. infantum* load, in agreement with other authors [2, 15].

## Conclusions

The high frequency, presence of active infection and similar *L. infantum* loads in the testis, epididymis, vulva and vagina of naturally infected dogs suggest the potential of venereal transmission of this parasite by both males and females in the area studied. In addition to venereal transmission, vertical transmission might be frequent in the canine population from the area studied since viable forms of *L. infantum* were commonly observed in the uterus.

## Acknowledgments

We thank the Municipal Health Department of Barra Mansa and the Central Laboratory of Public Health (LACEN) for their collaboration; Adilson Benedito de Almeida and Antonio Carlos da Silva from INI, Fiocruz, and Monique Paiva Campos from ICC, Fiocruz, for technical assistance, and Ricardo Baptista Schmidt from the Oswaldo Cruz Institute (IOC), Fiocruz, for processing the figures.

## Author Contributions

**Conceptualization:** Viviane Cardoso Boechat, Rodrigo Caldas Menezes.

**Data curation:** Viviane Cardoso Boechat, Artur Augusto Velho Mendes Júnior, Rodrigo Caldas Menezes.

**Formal analysis:** Viviane Cardoso Boechat, Raquel de Vasconcellos Carvalhaes de Oliveira, Rodrigo Caldas Menezes.

**Funding acquisition:** Sandro Antonio Pereira, Fabiano Borges Figueiredo, Rodrigo Caldas Menezes.

**Investigation:** Viviane Cardoso Boechat, Artur Augusto Velho Mendes Júnior, Shanna Araujo dos Santos, Luciana de Freitas Campos Miranda, Luiz Claudio Ferreira, Francisco das Chagas de Carvalho Rodrigues, Rayane Teles -de-Freitas, Rafaela Vieira Bruno, Rodrigo Caldas Menezes.

**Methodology:** Viviane Cardoso Boechat, Raquel de Vasconcellos Carvalhaes de Oliveira, Fernanda Nazaré Morgado, Rodrigo Caldas Menezes.

**Project administration:** Viviane Cardoso Boechat, Rodrigo Caldas Menezes.

**Resources:** Sandro Antonio Pereira, Rodrigo Caldas Menezes.

**Supervision:** Sandro Antonio Pereira, Rodrigo Caldas Menezes.

**Validation:** Viviane Cardoso Boechat, Raquel de Vasconcellos Carvalhaes de Oliveira, Rodrigo Caldas Menezes.

**Visualization:** Viviane Cardoso Boechat, Raquel de Vasconcellos Carvalhaes de Oliveira, Rodrigo Caldas Menezes.

**Writing – original draft:** Viviane Cardoso Boechat, Sandro Antonio Pereira, Fabiano Borges Figueiredo, Raquel de Vasconcellos Carvalhaes de Oliveira, Fernanda Nazaré Morgado, Rodrigo Caldas Menezes.

**Writing – review & editing:** Viviane Cardoso Boechat, Sandro Antonio Pereira, Artur Augusto Velho Mendes Júnior, Shanna Araujo dos Santos, Luciana de Freitas Campos Miranda, Fabiano Borges Figueiredo, Luiz Claudio Ferreira, Francisco das Chagas de Carvalho Rodrigues, Raquel de Vasconcellos Carvalhaes de Oliveira, Rayane Teles -de-Freitas, Rafaela Vieira Bruno, Fernanda Nazaré Morgado, Rodrigo Caldas Menezes.

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
