## [Decision Letter · Decision Letter 0]

12 Jun 2020

PONE-D-20-10750

High frequency, active infection and similar loads of Leishmania infantum in the genital tract of naturally infected male and female dogs

PLOS ONE

Dear Dr. Menezes,

Thank you for submitting your manuscript to PLOS ONE. After careful consideration, we feel that it has merit but does not fully meet PLOS ONE’s publication criteria as it currently stands. Therefore, we invite you to submit a revised version of the manuscript that addresses the points raised during the review process.

The manuscript highlights the *L. infantum* loads and inflammation in the genital tract of naturally infected dogs in an endemic area of Brazil by qPCR and IHC. Besides vertical transmission, it also suggests venereal transmission from both the sexes as also suggested by other studies. The positive aspect is the higher number of animals used in the present study; otherwise, there are no new findings compared to similar studies conducted in the past. 

The manuscript cannot be accepted as it presently stands. There are serious concerns that need to be addressed as elaborated in Reviewer’s comments. 

Further, the authors should incorporate the following suggestions:

Every method should be supported by a reference.Few references need to be updated. The authors have mentioned the results of other studies at several instances while discussing their findings, which is confusing. The exact values from other studies need not be mentioned every time.The dogs were naturally infected and of different ages. The authors should discuss why they found similar parasite loads in the different genital organs (testis, epididymis, vulva and vagina) and significantly low in prostrate and uterus, unlike other studies. They have mentioned the anatomic proximity of the organs. The title should be modified to bring out the crux of the study; it is too detailed at the moment.

We look forward to receiving your revised manuscript.

Kind regards,

Farhat Afrin, Ph.D.

Academic Editor

PLOS ONE

Additional Editor Comments:

The manuscript highlights the L. infantum loads and inflammation in the genital tract of naturally infected dogs in an endemic area of Brazil by qPCR and IHC. Besides vertical transmission, it also suggests venereal transmission from both the sexes as also suggested by other studies. The positive aspect is the higher number of animals used in the present study; otherwise, there are no new findings compared to similar studies conducted in the past.

The manuscript cannot be accepted as it presently stands. There are serious concerns that need to be addressed as elaborated in Reviewer’s comments.

Further, the authors should incorporate the following suggestions:

1) Every method should be supported by a reference.

2) Few references need to be updated.

3) The authors have mentioned the results of other studies at several instances while discussing their findings, which is confusing. The exact values from other studies need not be mentioned every time.

4) The dogs were naturally infected and of different ages. The authors should discuss why they found similar parasite loads in the different genital organs (testis, epididymis, vulva and vagina) and significantly low in prostrate and uterus, unlike other studies. They have mentioned the anatomic proximity of the organs.

5) The title should be modified to bring out the crux of the study; it is too detailed at the moment.

Journal Requirements:

Reviewers' comments:

Reviewer's Responses to Questions

**Comments to the Author**

1. Is the manuscript technically sound, and do the data support the conclusions?

Reviewer #1: Yes

Reviewer #2: Partly

Reviewer #3: Yes

2. Has the statistical analysis been performed appropriately and rigorously? 

Reviewer #1: Yes

Reviewer #2: I Don't Know

Reviewer #3: N/A

3. Have the authors made all data underlying the findings in their manuscript fully available?

Reviewer #1: Yes

Reviewer #2: No

Reviewer #3: Yes

4. Is the manuscript presented in an intelligible fashion and written in standard English?

Reviewer #1: Yes

Reviewer #2: Yes

Reviewer #3: Yes

5. Review Comments to the Author

Reviewer #1: The manuscript entitled, “High frequency, active infection and similar loads of

Leishmania infantum in the genital tract of naturally infected male and female dogs” focuses on to bring out a sensitive and with an elaborated sample number to address whether Leishmania infantum observed in the genital tract of naturally infected dogs of both the sex compared to couple of previous other reports in literature. As a first time, this manuscript comprehensively looks at the high frequency of infecting parasites in the testes, epididymides, vulva and vagina of the dogs and the possibility of venereal transmission of disease. In addition to qPCR they looked at histological alterations associated with parasitism in different organs of the dogs. In the area studied, in addition to venerable transmission, vertical transmission might be frequent in those dogs, as viable parasites were usually observed in the uterus. So they conclude that castration in such animals would only be a viable control measure of CVL.

It has been a well written article, with an appropriate sample number by following all needed ethical procedures and statistics. Couple of points need to be addressed.

1. In figure 2, The label at the x axis need to be corrected straight.

2. In Figure 3 and 4, the authors need to reconfirm on their scales. The scales first of all are not clearly visible. In figure 4, both d and f look written similar scale but magnifications could be different.

3. In the qPCR methods, the type of internal control or other that had been used for tissue amount uniformity for the comparison, needs to be specified.

4. In the methods of parasitological culture, information about what parameter was used to generate values observed in figure 1.

.

Reviewer #2: The manuscript “High frequency, active infection and similar loads of Leishmania infantum in the genital tract of naturally infected male and female dogs” brings some interesting information of CVL and reproduction in dogs. Its great merit is the number of animals used, higher than used in other works of the same field, and the findings of actively infected genital organs in dogs and bitches. However, I have some major commentaries:

1) The dogs and bitches were privately owned, not a controlled population. This is not a problem per se, but in this case other not controlled diseases can produce genital histophatological similar lesions, especially granulomatous inflammation, and may cause misinterpreted conclusions, like canine brucellosis, a silent disease worldwide distributed (see a recent published article in PlosOne - Eckstein C et al. (2020) Brucella ovis mutant in ABC transporter protects against Brucella canis infection in mice and it is safe for dogs. PLoS ONE 15(4):e0231893. https://doi.org/10.1371/ journal.pone.0231893);

2) The criterion for inclusion in this experiment was to have a serologically positive diagnosis of CVL, although the tests used are considered of good sensitivity and specificity (Ribeiro, VM et al. Performance of different serological tests in the diagnosis of natural infection by Leishmania infantum in dogs. Vet Parasitol, v.274, 108920, 2019. doi:10.1016/j.vetpar.2019.08.014), the parasitologically positive (bone marrow or lymph node smears) criteria would undoubtedly confirm that the animals were infected with Leishmania at the time of euthanasia, especially considering that you had 15% of dogs without clinical signs. The first works of the two last decades also used (not all) serological diagnosis. However, for description of the presence of Leishmania in organs, semen etc, and to characterize lesions, the goal of these articles, it was enough, but not to quantify parasitism and lesions already known and described. An option is to focus in analyze separately the animals with parasitological or PCR diagnosis in genital organs (69% of males and 64% of females; animals undoubtedly positive), considering that bone marrow, lymph node or other parasitological exam are now impossible. If you already have done this it was not clear to me, and needs to be clarified in Material and Methods and Results;

3) Differences among bitches under estrogenic milieu or not was showed by Magro et al. (2017), and this was not considered here. This is not prohibitive, but the inclusion of this aspect may be considered for enrichment of the results;

4) The Material and Methods should be carefully re-written, because some methods are not clear enough. For example, the use of positive/negative controls, and what did they do with double organs (testis and epididymis);

5) The Kappa index may be used to compare different diagnostic methods used for the same animal/organ;

6) Some discussions need to be rethought and improved. For example, about Leishmania progression after venereal contamination and others whose I will expose later.

I have some observations step by step, as follows:

Line 36: Put males before female, to keep the same order of presentation for the entire text, including abstract;

Lines 43-47: Change, for example, to “The high frequency, detection of active infection and similarity of L. infantum loads in the genital tract of infected males and females suggest that both sexes have potential for venereal transmission and bitches for vertical transmission of this parasite in the area studied”;

Lines 54-55: Include semen and smegma in the list, and put organs in a logical sequence, according to the anatomical sequence (i.e. testis, epididymis, prostate, glans penis, prepuce, smegma and semen). For smegma see Silva L.C. et al. Detection of Leishmania infantum in the smegma of infected dogs. Arq Bras Med Vet Zootec, v.66, p.731-736, 2014. doi: 10.1590/1678-41626610;

Lines 58-60: “In addition to providing a diagnostic result, this technique allows to determine the number of copies of the DNA target sequence” (include a reference);

Lines 70-72: Here may be interesting to include references of Leishmania venereal transmission from female to male in other species, like in humans;

Line 90: Consider substitute “… conducted between August 2015 and December 2016” using “from … to …”;

Lines 93-95: What was your intention when you classified dogs among different age ranges? You did not use this anytime in your analyses. However, estrous cycle stage (i.e. estrogenic – proestrus/estrus or non-estrogenic – diestrus/anestrus) may be interesting, as showed by Magro et al. (2017). You can do this using evaluation of the status of the vaginal epithelia, and also with the ovaries (if you have). If you find different estrous cycle stages, you can make comparisons between phases. If not, forget. This may enrich your article;

Lines 97-100: See initial commentaries. Also, include a reference for the validation of the serological tests used;

Line 106: That normative reference is too old (2006), 15 years ago. Please, see if there is a newest one;

Line 113: Hypotrophy is better;

Lines 113-115: This classification was used before? If yes, put reference;

Lines 119-120: Please, include reference;

Line 130: Maybe Drosophila needs to be in italics. I don’t know. Confirm;

Line 136: Exclude “collected”, and change tissues to organs. You evaluated organs, not tissues. This happens in several parts of the text. Change;

Line 147-149: I think this information needs to be referenced;

Line 156: For histopathology you put the tick of the specimen (line 164, good!). Put here too;

Lines 156-161: Did you use positive and negative controls? If yes, explain here. If not, it was not good!;

Lines 169-171: This classification was used before? If yes, put reference;

Line 174: “… variables were reported …;

Line 176: Change “… CVL), a positive result in the diagnostic …” to “… CVL), positive results in the diagnostic …”;

Line 178: Exclude “examined”;

Line 179: Spearman’s correlation test is used for non-parametric data. You said this later, but here you need to explain why you used Spearman test. The text needs to be reordered;

Line 181: I don’t know if non-normal distribution is a currently used terminology. Please, confirm that;

Line 182-184: Considering that, at the end of the paragraph you will put the significance, P < 0.05 here is unnecessary. Also, “significance of 95%” is better;

Line 182: Here is not clear if you compared all organs (male and female ones) among than, or if you compared male organs among than and female organs among than;

Lines 190-193: Exclude this phrase. It is too specific to be here, and you said this before (lines 116-120);

Lines 196-206: I would like to see the results for males and females separately here. Something like: “Clinical examination of the 70 dogs revealed clinical signs in 59 (84.3%), x males (x% of the males) and x females (x% of the females): thinness (x males and y females), ...”;

Lines 203-204: Here I am not sure if that animal was really infected at the time of euthanasia. I think that to exclude this animal from analyses would be better;

Line 207: Figure 1: I think that a table for these results can be better than the figure 1, with organs in lines and tests in columns. However, if figure was maintained, put organs in a logical order, testis before epididymis. Also, change testes to testis in the figure;

Lines 213-219: Testis and epididymis are double organs. What did you made with this? At least one organ positive was considered “testis” positive? The same for epididymis. It should be explained in material and methods. Also, differences among right and left organ was evaluated? This can be explored aiming to establish relationships between ipsilateral organs. This is applicable for all diagnostic procedures you used;

Line 216: Change testes to testis. Make a conference in all the text;

Lines 217-219: You may use Kappa index to compare different tests at the same organ. Consider this possibility to enrich your results;

Line 224: Clarify this in material and methods;

Lines 226-230: I am not yet convinced if comparisons among female and male organs are relevant. Why to compare epididymis and vulva, for example. Does it make sense? Comparisons among organs of the same gender make sense for me. I am not convinced but, if I am wrong, you may let these comparisons in the article;

Lines 242-243: Although table 1 show the results, I think that significant results have to be showed in the text, and the others generically classified as non-significant;

Lines 254-256: Here I am concerned about the possibility of lesions due to other causes but leishmaniasis;

Lines 295-296: The same observation for lines 242-243;

Lines 365-370: From witch part of the vagina specimens were collected? Vagina is very long! I think that this explanation is too much simple. The uterus is also near to the cranial vagina, and far from caudal vagina. It depends on the region of the vagina was sampled. It was included in your experimental protocol where vagina was sampled? I think this discussion should be excluded from the manuscript;

Lines 381-382: Considering an ascendant infection coming from penis, the next affected organ (from those exanimated) is prostate, second epididymis and, finally, testis. So, your argumentation did not support this hypothesis. Think on bacterial prostatitis in dogs. Most of them are from ascendant infection, without epididymitis and orchitis;

Line 384: … prepuce, penile glans and smegma. For smegma use the reference Silva L.C. et al. Detection of Leishmania infantum in the smegma of infected dogs. Arq Bras Med Vet Zootec, 66, 731-736, 2014. doi: 10.1590/1678-41626610;

Line 385: put uterine tubes before ovaries;

Lines 381-387: Your discussion is contradictory. For males the infection comes from the penis to testis/epididymis “jumping” prostate, but for females infection from vulva ascend “step by step”. It seems a “convenient” explanation to your findings, but weak. This hypothesis has to be rethought. The predilection of Leishmania for one or other organ probably is explained by other reasons, maybe the availability of parasites for venereal transmission, vulva/vagina to infect dogs, and penis/prepuce and semen (coming from testis/epididymis) to infect bitches. This is only an speculation, thinking on the predilection of the parasite to skin for vector contamination;

Lines 389-391: Despite these authors (2, 3 and 9) said this, it is not trough. The canine copula is not traumatic. The erection of the bulbus glandis inside vulva did not cause trauma in the vulva or vagina. It’s anecdotal. Unfortunately, I think that the authors used this argument trying to explain the bitch infection, but it is not sustainable by the mechanism of copula in dogs. Feline copula is traumatic, but not the canine copula. The literature about canine theriogenology did not classify dog's copula as traumatic. However, the friction of the contaminated surface of penis against female mucosae may be the way of Leishmania entrance inside female body. Silva et al. (2014) hypothesized that smegma may be the mainly source of Leishmania for female contamination, although not proved. This discussion and hypothesis also has to be rethought;

Lines 396-399: As discussed before, I think it is plausible. Also, in the experiment where venereal transmission was showed (Silva et al., 2009), the infected dog and susceptive bitch were put together to copulate, and authors conclude that semen was the vehicle of Leishmania to the bitch. The contact of dog and bitch other than semen and male/female genitalia contact could be the source of infection, including penile-prepuce-smegma and vulva-vagina, as far as due to oral infection with contaminated fluids, if possible (not reported yet);

Lines 406-408: I think that this speculative suggestion should be accompanied by something like this: "However, it needs to be confirmed …". Please, include;

Lines 422-424: Assis et al. (2010) found poor semen characteristics in dogs with CVL, but attributed the low semen quality probably to epididymal affection. You and others found epididymitis associated to Leishmania presence in the organ. I think this could be discussed here, besides of the testis discussion, as epididymis affection seems to be more important than testis affection lowering semen quality (Assis V.P. et al. Dogs with Leishmania chagasi infection have semen abnormalities that partially revert during 150 days of Allopurinol and Amphotericin B therapy. Anim. Reprod. Sci., v.117, p.183-186, 2010. doi: 10.1016/j.anireprosci.2009.03.003). Labat et al. (2010) also showed semen abnormalities in dogs with CVL, however, they did not explained clearly how the dogs were diagnosed for CVL (Labat E. et al. Qualidade espermática de sêmen de cães naturalmente infectados por Leishmania sp. Arq Bras Med Vet Zootec, v.62, p.609-614, 2010. doi: 10.1590/S0102-09352010000300016). Consider discussing this;

Lines 437-438: It cannot be placed in conclusion, but you can put at the end of the discussion.

Reviewer #3: The manuscript provides sounding results concerning the presence of Leishmania parasite in the genital tract of male and female dogs. The manuscript is relevant with supporting results. Thus, it should be accepted for publication.

6. PLOS authors have the option to publish the peer review history of their article (what does this mean?). If published, this will include your full peer review and any attached files.

Reviewer #1: Yes: Angamuthu Selvapandiyan

Reviewer #2: No

Reviewer #3: Yes: Claudio Vieira da Silva

---

## [Author Response · Author response to Decision Letter 0]

28 Jul 2020

RESPONSE TO ACADEMIC EDITOR

PONE-D-20-10750

High frequency, active infection and similar loads of Leishmania infantum in the genital tract of naturally infected male and female dogs

PLOS ONE

Dear Dr. Menezes,

Thank you for submitting your manuscript to PLOS ONE. After careful consideration, we feel that it has merit but does not fully meet PLOS ONE’s publication criteria as it currently stands. Therefore, we invite you to submit a revised version of the manuscript that addresses the points raised during the review process.

Additional Editor Comments:

The manuscript highlights the L. infantum loads and inflammation in the genital tract of naturally infected dogs in an endemic area of Brazil by qPCR and IHC. Besides vertical transmission, it also suggests venereal transmission from both the sexes as also suggested by other studies. The positive aspect is the higher number of animals used in the present study; otherwise, there are no new findings compared to similar studies conducted in the past.

The manuscript cannot be accepted as it presently stands. There are serious concerns that need to be addressed as elaborated in Reviewer’s comments.

Further, the authors should incorporate the following suggestions:

Further, the authors should incorporate the following suggestions:

1) Every method should be supported by a reference.

Response

The references for the serological methods were included. The reference included for the rapid dual-path platform (TR DPP®) assay was Schubach et al. (2014) [22] and the reference included for enzyme immunoassay (ELISA) was Arruda et al. (2016) [23] (Line 105). In addition, two references were included for the diagnosis criteria recommended by the Brazilian Ministry of Health to consider a dog as infected by L. infantum and to submit it for euthanasia as a control measure for ZVL. These references were Ribeiro et al. (2019) [24] and Brasil (2011) [25].

References (lines 624-636):

22-Schubach EYP, Figueiredo FB, Romero GAS. Accuracy and reproducibility of a rapid chromatographic immunoassay for the diagnosis of canine visceral leishmaniasis in Brazil. Trans R Soc Trop Med Hyg. 2014;108: 568–574.

23-Arruda MM, Figueiredo FB, Marcelino AP, Barbosa JR, Werneck GL, Noronha EF, et al. Sensitivity and specificity of parallel or serial serological testing for detection of canine Leishmania infection. Mem Inst Oswaldo Cruz. 2016; 111: 168-173.

24-Ribeiro VM, Miranda JB, Marcelino AP, Andrade HM, Reis IA, Cardoso MS, et al. Performance of different serological tests in the diagnosis of natural infection by Leishmania infantum in dogs. Vet Parasitol. 2019; 274: 108920.

25-Brasil. Esclarecimento sobre substituição do protocolo diagnóstico da leishmaniose visceral canina (LVC); Nota Técnica Conjunta n ° 01/2011 - CGDT-CGLAB / Devit / SVS / MS. 2011. Available: http://www.sgc.goias.gov.br/upload/arquivos/2012-05/nota-tecnica-no.-1-2011_cglab_cgdt1_lvc.pdf. Accessed 25 June 2020.

The following reference for the guidelines of the Federal Council on Veterinary Medicine of Brazil was also included: CFMV (2013) [28] (line 130)

Reference (lines 643-646): 

28-Conselho Federal de Medicina Veterinária. Guia brasileiro de boas práticas para eutanásia em animais- conceitos e procedimentos recomendados. 2013. Available: http://portal.cfmv.gov.br/uploads/files/Guia%20de%20Boas%20Pr%C3%A1ticas%20para%20Eutanasia.pdf.pdf. Accessed 25 June 2020.

A reference for the parameter used for a positive diagnosis in the parasitological culture was included: Madeira et al. (2009) [30] (line 147). The link for the register of the detailed protocol of parasitic isolation in culture was also included: https://dx.doi.org/10.17504/protocols.io.22tggen.

Reference (lines 649-651)

30-Madeira MF, Figueiredo FB, Pinto AGS, Nascimento LD, Furtado M, Mouta-Confort E, et al. Parasitological diagnosis of canine visceral leishmaniasis: is intact skin a good target? Res Vet Sci. 2009;87: 260-262.

In the methodology of qPCR, a reference for expression of the L. infantum load as the natural logarithm of the number of parasite genome equivalents (gEq)/ng was included: Oliveira et al. (2017) [32] (line 169).

Reference (lines 654-657)

32- Oliveira VDC, Boechat VC, Mendes Junior AAV, Madeira MF, Ferreira LC, Figueiredo FB, et al. Occurrence of Leishmania infantum in the central nervous system of naturally infected dogs: Parasite load, viability, co-infections and histological alterations. PLoS One. 2017;12:e0175588.

In the statistical analysis, a reference for Cohen’s kappa (k) statistic was included: Landis and Koch, 1977 [34] in the line 232.

Reference (lines 161-162)

34- Landis JR, Koch GG. The measurement of observer agreement for categorical data. Biometrics. 1977;33: 159-174.

The other methods described in the manuscript were supported by a reference:

Line 29. Fixation in 10% buffered formalin and embedded in paraffin [29].

29-Carson FL, Cappellano CH. Histotechnology: a self-instructional text. 4th ed. Chicago: ASCP Press; 2015. 

Lines 145-150. Parasitological culture and identification of Leishmania species [30, 31].

30-Madeira MF, Figueiredo FB, Pinto AGS, Nascimento LD, Furtado M, Mouta-Confort E, et al. Parasitological diagnosis of canine visceral leishmaniasis: is intact skin a good target? Res Vet Sci. 2009;87: 260-262.

31-Cupolillo E, Grimaldi Jr G, Momen H. A general classification of New World Leishmania using numerical zymotaxonomy. Am J Trop Med Hyg. 1994;50: 296-311.

Lines 161, 169 and 172. Singleplex qPCR for the diagnosis and quantification of Leishmania infantum load [32].

32-Oliveira VDC, Boechat VC, Mendes Junior AAV, Madeira MF, Ferreira LC, Figueiredo FB, et al. Occurrence of Leishmania infantum in the central nervous system of naturally infected dogs: Parasite load, viability, co-infections and histological alterations. PLoS One. 2017;12:e0175588.

Immunohistochemistry [11], line 185.

11-Boechat VC, Mendes Junior AAV, Madeira MF, Ferreira LC, Figueiredo FB, Rodrigues FCC, et al. Occurrence of Leishmania infantum and associated histological alterations in the genital tract and mammary glands of naturally infected dogs. Parasitol Res. 2016;115:2371-2379.

Histopathology [29], line 198.

29-Carson FL, Cappellano CH. Histotechnology: a self-instructional text. 4th ed. Chicago: ASCP Press; 2015. 

2) Few references need to be updated. 

Response

Lines 55 and 115. The reference Brasil (2006) was updated to Brasil (2016) [1], as shown below:

References (lines 563-565)

1. Brasil. Ministério da Saúde. Secretaria de Vigilância em Saúde. Departamento de Vigilância das Doenças Transmissíveis. Manual de vigilância, prevenção e controle de zoonoses: normas técnicas e operacionais. Brasília: Ministério da Saúde; 2016.

Lines 691-692. The reference Jubb et al. (2007) was corrected and updated to Maxie (2016) [44].

44-Maxie MG. Jubb, Kennedy, and Palmer’s pathology of domestic animals. 6th. ed. St. Louis: Elsevier; 2016.

Other references different from that mentioned above were included:

References

10-Silva LC, Assis VP, Ribeiro VM, Tafuri WL, Toledo Júnior JC, Silva SO. Detection of Leishmania infantum in the smegma of infected dogs. Arq Bras Med Vet Zootec. 2014;66:731-736. (lines 587-589).

Cited in the lines 59, 482, 491.

17-Galluzzi L, Ceccarelli M, Diotallevi A, Menotta M, Magnani M. Real-time PCR applications for diagnosis of leishmaniasis. Parasit Vectors. 2018;11:273. (lines 610-611).

19-Rosypal AC, Lindsay DS. Non-sand fly transmission of a North American isolate of Leishmania infantum in experimentally infected BALB/c mice. J Parasitol. 2005; 91:1113-1115. (lines 614-616).

Cited in the line 63. 

46- Hedger MP. The Immunophysiology of Male Reproduction. In: Plant TM, Zeleznick A. Zachary JF, McGavin MD, editors. Knobil and Neill’s Physiology of Reproduction. 4th ed. San Diego: Elsevier; 2015. p. 805-892. (lines 695-697).

Cited in the line 466. 

48-Hollett RB. Canine brucellosis: outbreaks and compliance. Theriogenology. 2006;66:575–587. (lines 700-701).

Cited in the line 524. 

50-Assis VP, Ribeiro VM, Rachid MA, Castro ACS, Valle GR. Dogs with Leishmania chagasi infection have semen abnormalities that partially revert during 150 days of allopurinol and amphotericin B therapy. Anim Reprod Sci. 2010;117: 183–186. (lines 704-706).

Cited in the line 536.

51-Labat E, Carreira JT, Matsukuma BH, Martins MTA, Lima VMF, Bomfim SRM, et al. Semen quality of dogs naturally infected by Leishmania sp. Arq Bras Med Vet Zootec. 2010;62: 609-614. (lines 707-709).

Cited in the line 536. 

3) The authors have mentioned the results of other studies at several instances while discussing their findings, which is confusing. The exact values from other studies need not be mentioned every time.

Response

The first three paragraphs of the discussion, in which the exact values from other studies were described, were reviewed according to the recommendations. We haven’t found repetition of results from the same authors. In the first paragraph, we have described the results of the general frequency of L. infantum infection in the genital tract of male and female dogs, without comparing the different organs. In the second and third paragraphs, we mentioned the results from other studies about the frequency of L. infantum infection in the different genital organs of male and female dogs.

However, in this revision, we reduced the text in the lines 392-394 in order to make it clear, as shown below:

The text “….as the frequency of amastigote forms of this parasite was low and no specific inflammatory alterations were observed in the genital organs [12]. These authors [12] detected amastigote forms of Leishmania only in the vulva, with a frequency of 20% by IHC and HP.” was replaced with the following text:

“…since amastigote forms of this parasite were detected only in the vulva, with a low frequency (20%) by IHC and HP, and no specific inflammatory alterations were observed in the genital organs [6].”

4) The dogs were naturally infected and of different ages. The authors should discuss why they found similar parasite loads in the different genital organs (testis, epididymis, vulva and vagina) and significantly low in prostrate and uterus, unlike other studies. They have mentioned the anatomic proximity of the organs. 

Response

In case of males, the similar and positive correlated parasite loads in the testis and epididymis can be explained by the anatomic proximity and communication between these organs. In addition, the lower frequency and load of L. infantum in the prostate compared to the epididymis and testis in in this study and in others [4, 11, 15], may be due to a more efficient innate and adaptive immunity against L. infantum in this organ. An important mechanism of innate immunity that protects the prostate from infections is the blood-prostate barrier [Fulmer and Turner, 2000]. A temperature lower than the body temperature and reduced immune responsiveness in the testis, as well as the lack of a developed mucosal immune system in the epididymis [45, 46], may predispose these organs to the infection with L. infantum.

In case of females, the similar and positive correlated parasite loads in the vulva and caudal vaginal can be explained by the anatomic proximity and communication between these organs. In addition, the infection would spread via the hematogenous or lymphatic route from the vulva and vagina to the uterus, uterine tubes and ovaries. This hypothesis may explain the lower frequency and parasite load in the uterus in the present study and in the uterus, uterine tubes and ovaries in the study of Boechat et al. [11] when compared to the external genital organs. 

However, we included in the discussion that future experimental studies on the kinetics of genital tract infection with L. infantum in dogs are necessary to confirm all of the hypotheses raised in the present study.

The discussion was reviewed and these hypotheses were included in the discussion according to recommendations of the reviewer #2, as indicated below:

Lines 457-466. 

“….The positive correlation between parasite load in the testis and epididymis and in the vulva and caudal vagina adjacent to the vulva observed in the present study can be explained by the anatomic proximity and communication between these organs [43, 44]. The lower frequency and load of L. infantum in the prostate compared to the epididymis and testis observed in this study and in others [4, 11, 15] may be due to a more efficient innate and adaptive immunity against L. infantum in this organ. An important mechanism of innate immunity that protects the prostate from infections is the blood-prostate barrier [42, 45]. A temperature lower than the body temperature and reduced immune responsiveness in the testis, as well as the lack of a developed mucosal immune system in the epididymis [45, 46], may predispose these organs to the infection with L. infantum.”

Lines 480-487.

“In males, the infection would then spread via the hematogenous or lymphatic route from the penis, which shows a high frequency of this parasite in the prepuce, glans penis and smegma [10, 11, 13, 15], to the testis, epididymis and prostate. In females, the infection would spread via the hematogenous or lymphatic route from the vulva and vagina to the uterus, uterine tubes and ovaries. This hypothesis may explain the lower frequency and parasite load in the uterus observed in the present study and in the uterus, uterine tubes and ovaries in the study of Boechat et al. [11] when compared to the external genital organs.” 

Lines 500-502. “However, future experimental studies on the kinetics of genital tract infection by L. infantum in dogs are necessary to confirm all of these hypotheses raised in the present study.”

5) The title should be modified to bring out the crux of the study; it is too detailed at the moment.

Response

The title was modified as recommended. The new title is “Frequency, active infection and load of Leishmania infantum and associated histological alterations in the genital tract of male and female dogs”.

RESPONSE TO REVIEWER #1

Reviewer #1: The manuscript entitled, “High frequency, active infection and similar loads of Leishmania infantum in the genital tract of naturally infected male and female dogs” focuses on to bring out a sensitive and with an elaborated sample number to address whether Leishmania infantum observed in the genital tract of naturally infected dogs of both the sex compared to couple of previous other reports in literature. As a first time, this manuscript comprehensively looks at the high frequency of infecting parasites in the testes, epididymides, vulva and vagina of the dogs and the possibility of venereal transmission of disease. In addition to qPCR they looked at histological alterations associated with parasitism in different organs of the dogs. In the area studied, in addition to venerable transmission, vertical transmission might be frequent in those dogs, as viable parasites were usually observed in the uterus. So they conclude that castration in such animals would only be a viable control measure of CVL.

It has been a well written article, with an appropriate sample number by following all needed ethical procedures and statistics. Couple of points need to be addressed.

1. In figure 2, The label at the x axis need to be corrected straight.

Response

Figure 1 (previous Figure 2). In the y axis, the legend was written horizontally, as recommended. The legend in the x axis was straight.

2. In Figure 3 and 4, the authors need to reconfirm on their scales. The scales first of all are not clearly visible. In figure 4, both d and f look written similar scale but magnifications could be different.

Response

The scales of Figures 2 and 3 (Figures 3 and 4 in the first version of the manuscript) were revised and corrected. The thickness of the bars and size of letters of all figures were increased in order to make the scales more visible.

3. In the qPCR methods, the type of internal control or other that had been used for tissue amount uniformity for the comparison, needs to be specified.

Response

Tissue fragments ≤25 mg were used. This specification was included in the line 156.

4. In the methods of parasitological culture, information about what parameter was used to generate values observed in figure 1.

The parasitological culture was considered positive if promastigote forms of Leishmania grew in the culture medium up to 30 days and were visualized by conventional optical microscopy [30]. The detailed protocol of parasite isolation in culture is registered at https://dx.doi.org/10.17504/protocols.io.22tggen. This text was included in the lines 145-149.

RESPONSE TO REVIEWER #2

Reviewer #2: The manuscript “High frequency, active infection and similar loads of Leishmania infantum in the genital tract of naturally infected male and female dogs” brings some interesting information of CVL and reproduction in dogs. Its great merit is the number of animals used, higher than used in other works of the same field, and the findings of actively infected genital organs in dogs and bitches. However, I have some major commentaries:

1) The dogs and bitches were privately owned, not a controlled population. This is not a problem per se, but in this case other not controlled diseases can produce genital histophatological similar lesions, especially granulomatous inflammation, and may cause misinterpreted conclusions, like canine brucellosis, a silent disease worldwide distributed (see a recent published article in PlosOne - Eckstein C et al. (2020) Brucella ovis mutant in ABC transporter protects against Brucella canis infection in mice and it is safe for dogs. PLoS ONE 15(4):e0231893. https://doi.org/10.1371/ journal.pone.0231893);

Response

We agree that Brucella canis infection is a differential histological diagnosis of L. infantum infection in the genital tract of dogs, as both pathogens may cause granulomatous inflammation in these organs. However, our results indicate that the histological alterations were associated with active L. infantum infection. The reasons are that: 100% females and 98% males were positive for L. infantum in the genital tract; active infection in the genital tract was confirmed in 69% of males and 64% of females; the frequency of inflammatory reaction was much higher in organs that tested positive by the parasitological methods compared to those in which no parasites were detected by these tests; granulomatous inflammation, which is expected as a response to the infection of different organs with L. infantum in dogs, predominated in all organs, except for the prostate. Even in the 19 dogs in which histological alterations were not associated with the detection of L. infantum by parasitological methods, the inflammatory reaction was probably associated with this parasite. The reason is that DNA of L. infantum was detected in the affected genital organs of these dogs. In addition, in these 19 dogs, the inflammation may have been caused indirectly by L. infantum via immunocomplex deposition [47]. The predominance of a non-granulomatous inflammatory reaction in the prostate, as also observed in other studies involving dogs with CVL, might be explained by the lower parasite load and lower frequency of L. infantum-positive specimens compared to the other organs examined. 

Following your comments, we have reviewed the results and discussion, as shown below:

Results

Lines 329-330. 

“In these 19 dogs, the inflammatory infiltrate observed in the genital tract organs was associated with the detection of L. infantum DNA in these organs.”

Discussion

Lines 518-525

 “Even in the 19 dogs in which histological alterations were not associated with the detection of L. infantum by parasitological methods, the inflammatory reaction was probably associated with this parasite, as L. infantum DNA was detected in the affected genital organs of these dogs. In addition, in these 19 dogs, the inflammation may have been caused indirectly by L. infantum via immunocomplex deposition [47]. However, the bacterium Brucella canis may also cause granulomatous inflammation in the genital tract of dogs [48]. Although unlikely, its participation in this type of inflammation in the present study should therefore not be completely ruled out.”

The reference of Hollet et al. (2006) [48] was included.

48-Hollett RB. Canine brucellosis: outbreaks and compliance. Theriogenology. 2006;66:575–587.

2) The criterion for inclusion in this experiment was to have a serologically positive diagnosis of CVL, although the tests used are considered of good sensitivity and specificity (Ribeiro, VM et al. Performance of different serological tests in the diagnosis of natural infection by Leishmania infantum in dogs. Vet Parasitol, v.274, 108920, 2019. doi:10.1016/j.vetpar.2019.08.014), the parasitologically positive (bone marrow or lymph node smears) criteria would undoubtedly confirm that the animals were infected with Leishmania at the time of euthanasia, especially considering that you had 15% of dogs without clinical signs. The first works of the two last decades also used (not all) serological diagnosis. However, for description of the presence of Leishmania in organs, semen etc, and to characterize lesions, the goal of these articles, it was enough, but not to quantify parasitism and lesions already known and described. An option is to focus in analyze separately the animals with parasitological or PCR diagnosis in genital organs (69% of males and 64% of females; animals undoubtedly positive), considering that bone marrow, lymph node or other parasitological exam are now impossible. If you already have done this it was not clear to me, and needs to be clarified in Material and Methods and Results;

Response

In the present study, 100% of females and 98% of males were undoubtedly infected with Leishmania infantum by at least one of the diagnostic techniques used (parasitological methods and qPCR). The combined parasitological techniques (culture, HP, and IHC) detected active Leishmania infection in the genital tract of 69% (n = 31) of males and 64% (n = 16) of females. The qPCR technique detected L. infantum DNA in at least one genital tract sample in 98% of males (n = 44) and 96% of females (n = 24). The only negative animal in all assays was a male without clinical signs. Therefore, according to the recommendations we excluded this animal from the analyses that compare the L. infantum load in the organs of the genital tract according to the clinical signs (Table 3) and intensity of inflammatory infiltrate in these organs (Table 5). The negative male was excluded of this comparisons and the values recalculated; however, there was no alterations in the results. In order to clarify the dogs included in these analysis, with the exclusion of the male that was only seropositive, we did the following modifications in the text:

Methods

The following text was included: 

Lines 174-179.

“For the comparison of parasite load between the different organs of the male and female genital tract, only dogs positive for L. infantum DNA in the genital tract by qPCR were analyzed. In turn, for the comparison of parasite load between the different organs of the male and female genital tract according to the clinical signs of the dogs and intensity of inflammatory infiltration in these organs, only dogs that tested positive for L. infantum by qPCR or parasitological methods were analyzed.”

Results

Legend of Fig 1. Lines 304-305. The text was reviewed and the number of dogs was changed by 68 due to the exclusion of the male dog negative in all tests. The reviewed text is written below:

“…genome equivalents/nanogram DNA (gEq/ng) in testis (a pool of right and left organs), epididymis (a pool of right and left organs), prostate, vulva, vagina, and uterus of 68 dogs positive for L. infantum DNA in the genital tract by qPCR.…”

Table 3 (lines 317-323). One male dog was excluded with no alterations in the values of parasite load. Therefore, the following alterations in the legend were done:

The number of dogs with no or few clinical signs was corrected. “N=45” was replaced with “N=44”.

Lines 323. The following text was included:

“dOne male dog was excluded because it tested negative for L. infantum by qPCR and parasitological methods.”

Table 5 (lines 372-379). One male dog was excluded with no alterations in the values of parasite load. Therefore, the following alterations in the legend were done:

The number of male dogs was corrected. “N=45” was replaced with “N=44”.

The following text was included (line 377): 

“bOne male dog was excluded because it tested negative for L. infantum by qPCR and parasitological methods.”

However, for the comparison of the frequency of histological alterations, the dogs had already been analyzed separately according to the parasitological diagnosis in the genital tract. One group was composed by dogs that tested positive and the other group was composed by dogs that tested negative for Leishmania by the parasitological methods (culture, HP or IHC). The aim of this comparison was to verify the association between lesion with active infection by Leishmania. In order to clarify the analysis of histological alteration, the following text was included in methods, sub item histopathology:

Lines 204-208. “The frequency of histological alterations in the genital tract was compared between the group of dogs that tested positive and the group of dogs that tested negative for Leishmania by the parasitological methods (parasitological culture, HP or IHC) in these organs. This comparison was done to verify the association between lesions with active Leishmania infection.”

The reason to use as inclusion criteria, dogs that have serologically positive diagnosis of CVL and not necessarily a parasitological or qPCR positive test, was to analyze the population of dogs in Brazil that are submitted to euthanasia as control measure of zoonotic visceral leishmaniasis, according to criteria of Brazilian Ministry of Health. In this criteria, the dog is considered positive for L. infantum and recommended for euthanasia if it is positive in both TR DPP® and ELISA (BioManguinhos, Fiocruz, Rio de Janeiro, Brazil). When designed the study, we considered important to analyze the frequencies of L. infantum in the genital tract of this group of dogs in order to know if they have potential for venereal transmission of this parasite in the area studied. In the present study, the detection of L. infantum parasitism in almost a 100% of the genital tract of males and females seropositive for this parasite, confirms the high specificity already described for the combination of TR DPP® and ELISA tests [24]. Therefore, including only dogs with confirmed infection by L. infantum by parasitological methods and/or qPCR, would have caused a bias in the results of the present study. For this reason, we did not exclude the male dog that was negative for L. infantum by qPCR or parasitological methods from the study. In order to clarify the design of study and aims, according to the justification above, the following modifications in the text were done:

Abstract

Lines 30-31. The text of the aims: “...in the genital tract of naturally infected male and female dogs from the same endemic area,...” was replaced with “genital tract of male and female dogs seropositive for this parasite,…”.

Line 33. We replaced “from an endemic area in Brazil” with “from the same endemic area in Brazil”.

Introduction

Lines 91 to 92. The text of the aims: “…of the genital tract of naturally infected male and female dogs from the same endemic area.” was replaced with “…in different organs of the genital tract, in male and female dogs seropositive for this parasite from the same endemic area.”

Methods

Lines 104-109. The following text was included:

“All dogs were privately owned and tested seropositive for anti-Leishmania antibodies by the rapid dual-path platform (TR DPP®) assay [22] and enzyme immunoassay (ELISA) [23], both produced by BioManguinhos (Fiocruz, Rio de Janeiro, Brazil). These two serological tests and diagnostic criteria were used because they are accurate [24] and are recommended by the Brazilian Ministry of Health for considering a dog to be infected with L. infantum and for submitting it to euthanasia as a control measure for CVL [25].”

Four new references were included, as shown below (lines 624-636):

22-Schubach EYP, Figueiredo FB, Romero GAS. Accuracy and reproducibility of a rapid chromatographic immunoassay for the diagnosis of canine visceral leishmaniasis in Brazil. Trans R Soc Trop Med Hyg. 2014;108: 568–574.

23-Arruda MM, Figueiredo FB, Marcelino AP, Barbosa JR, Werneck GL, Noronha EF, et al. Sensitivity and specificity of parallel or serial serological testing for detection of canine Leishmania infection. Mem Inst Oswaldo Cruz. 2016; 111: 168-173.

24-Ribeiro VM, Miranda JB, Marcelino AP, Andrade HM, Reis IA, Cardoso MS, et al. Performance of different serological tests in the diagnosis of natural infection by Leishmania infantum in dogs. Vet Parasitol. 2019; 274: 108920.

25-Brasil. Esclarecimento sobre substituição do protocolo diagnóstico da leishmaniose visceral canina (LVC); Nota Técnica Conjunta n ° 01/2011 - CGDT-CGLAB / Devit / SVS / MS. 2011. Available: http://www.sgc.goias.gov.br/upload/arquivos/2012-05/nota-tecnica-no.-1-2011_cglab_cgdt1_lvc.pdf. Accessed 25 June 2020.

Results

Table 4. Legend. Line 356. The text “naturally infected by” was replaced with “seropositive for”.

Discussion

Lines 401-404. The following text was included: 

“In the present study, the detection of L. infantum parasitism in almost 100% of the genital tract of males and females seropositive for this parasite confirms the high specificity already described for the combination of TR DPP® and ELISA tests [24].”

The reference Ribeiro et al. (2019) [24] was added in the lines 630-632.

3) Differences among bitches under estrogenic milieu or not was showed by Magro et al. (2017), and this was not considered here. This is not prohibitive, but the inclusion of this aspect may be considered for enrichment of the results;

Response

Unfortunately, we don’t have these results.

4) The Material and Methods should be carefully re-written, because some methods are not clear enough. For example, the use of positive/negative controls, and what did they do with double organs (testis and epididymis);

Response

We have re-written the Methods in order to clarify the use to the use of positive/negative controls and what did they do with double organs (testis and epididymis).

The following texts were included:

Methods.

Sample collection. Lines 136-140.

 “In the case of double organs, one fragment of the right and left testis and epididymis each was collected and examined separately by HP and IHC. For parasitological culture and qPCR, a pool of samples containing one fragment of the right and left testis and a pool of samples containing one fragment of the right and left epididymis were examined.”

Immunohistochemistry. Lines 188-193. 

 “Histological sections of organs intensely parasitized with amastigote forms of Leishmania were incubated with non-immune homologous serum as negative control and with polyclonal rabbit anti-Leishmania serum as positive control. The testis and epididymis were considered positive for Leishmania if amastigote forms of this parasite were detected in at least one fragment of the right or left organ.”

Histopathology. Lines 209-216. 

“The testis and epididymis were considered positive for Leishmania if amastigote forms of this parasite were detected in at least one fragment of the right or left organ. For calculation of the frequency of each histological alteration in the testis and epididymis, the histological alteration was considered present if detected in at least one fragment of the right or left organ. The intensity of inflammatory infiltration was classified in the organ (right or left) with the most intense inflammatory infiltrate. In addition, the frequencies of positivity for amastigote forms of L. infantum and of inflammatory infiltrates associated with amastigote forms of this parasite in the right and left testis and epididymis were evaluated.”

Singleplex qPCR for the diagnosis and quantification of Leishmania infantum load. Lines 164-166.

 “The DNA of 1 x 105 promastigote forms of L. infantum obtained by parasitological culture was used as positive control and ultrapure water as negative control.”

Results

Legend of Table 1. Lines 259-262.

“aA pool of samples containing one fragment of the right and left testis and a pool of samples containing one fragment of the right and left epididymis were examined.”

“bThe testis and epididymis were considered positive for Leishmania if amastigote forms of this parasite were detected in at least one fragment of the right or left organ.”

Legend of Fig 1. Lines 303-305. 

“…genome equivalents/nanogram DNA (gEq/ng) in testis (a pool of right and left organs), epididymis (a pool of right and left organs), prostate, vulva, vagina, and uterus of 68 dogs positive for L. infantum DNA in the genital tract by qPCR.”

Legend of Table 4. Line 361.

 “bThe histological alteration was considered present if detected in at least one fragment of the right or left organ.”

Legend of Table 5. Lines 378-379.

“cThe intensity of inflammatory infiltration was classified in the organ (right or left) with the most intense inflammatory infiltrate.”

5) The Kappa index may be used to compare different diagnostic methods used for the same animal/organ;

Response

The level of agreement between the results of the diagnostic techniques (positive/negative) for detection of Leishmania infection in all organs of the genital tract was assessed by calculating Cohen’s kappa (k) statistic, as recommended. Therefore, the following modifications in the text of manuscript were done:

Lines 229-232. Statistical analysis. 

The following text was included: “The level of agreement between the results of the diagnostic techniques (positive/negative) for detection of Leishmania infection in all organs of the genital tract was assessed by calculating Cohen’s kappa (k) statistic, considering the classification proposed by Landis and Koch [34].”

Lines 265-267. Results. The following text was included: “Table 2 shows the level of agreement between the results of the diagnostic techniques for detection of Leishmania infection in all organs of the genital tract using Cohen’s kappa statistic.”

Lines 269-275. Results. A Table with the results of kappa index (Table 2) was included.

Lines 418-426. Discussion. The following text was included: “In the present study, the level of agreement between the parasitological techniques was better than the level of agreement between qPCR and parasitological techniques in all organs examined, except for the prostate. These results can be explained by the differences in the frequencies of positivity for Leishmania. These differences were smaller between parasitological techniques and greater when qPCR and the parasitological techniques were compared. The latter are considered less sensitive than qPCR [32]. Among the genital tract organs, the lowest level of agreement between the diagnostic techniques that could be compared was observed for the prostate. This finding is related to the low frequency of positivity for Leishmania in this organ by parasitological techniques. 

6) Some discussions need to be rethought and improved. For example, about Leishmania progression after venereal contamination and others whose I will expose later.

I have some observations step by step, as follows:

Line 36: Put males before female, to keep the same order of presentation for the entire text, including abstract;

Response

We put males before female for the entire text, as indicated below:

Abstract. Line 37. The text “Twenty-five (100%) females and 44 (98%) males were positive for L. infantum…” was replaced with “Forty-four (98%) males and 25 (100%) females…”

Results. Line 250. The text “ In the genital tract of the 25 females and 45 males, 100% of females and 98% of males” was replaced with “ In the genital tract of the 45 males and 25 females, 98% of males and 100% of females…”, in order to follow the same recommendation.

Lines 43-47: Change, for example, to “The high frequency, detection of active infection and similarity of L. infantum loads in the genital tract of infected males and females suggest that both sexes have potential for venereal transmission and bitches for vertical transmission of this parasite in the area studied”;

Response

Abstract. 

The phrase was changed, as recommended.

Lines 45-47. “The high frequency, detection of active infection and similarity of L. infantum loads in the genital tract of infected males and females suggest the potential of venereal transmission of this parasite by both sexes and of vertical transmission by females in the area studied.

Lines 54-55: Include semen and smegma in the list, and put organs in a logical sequence, according to the anatomical sequence (i.e. testis, epididymis, prostate, glans penis, prepuce, smegma and semen). For smegma see Silva L.C. et al. Detection of Leishmania infantum in the smegma of infected dogs. Arq Bras Med Vet Zootec, v.66, p.731-736, 2014. doi: 10.1590/1678-41626610;

Response

The scrotum, semen and smegma were included in the list and organs were written in a logical sequence, as recommended. The reference for smegma of Silva et al. (2014) [10] was included.

Therefore, the following text was included:

Introduction. Lines 57-59. “…in the testis, epididymis, prostate, glans penis, prepuce, scrotum, smegma, semen, vulva, vagina, vaginal secretion, placenta, uterus, uterine tubes, and ovaries [2, 4, 6-16].”

Lines 58-60: “In addition to providing a diagnostic result, this technique allows to determine the number of copies of the DNA target sequence” (include a reference);

Response

Line 63. We included the following reference [17]:

17-Galluzzi L, Ceccarelli M, Diotallevi A, Menotta M, Magnani M. Real-time PCR applications for diagnosis of leishmaniasis. Parasit Vectors. 2018;11:273. (lines 610-61)

Lines 70-72: Here may be interesting to include references of Leishmania venereal transmission from female to male in other species, like in humans;

Response

A reference of an experimental study that suggests Leishmania venereal transmission from female to male in mice was included: Rosypal et al. (2005) [19]. References of Leishmania venereal transmission from female to male in humans were not found.

Reference (lines 614-616):

19-Rosypal AC, Lindsay DS. Non-sand fly transmission of a North American isolate of Leishmania infantum in experimentally infected BALB/c mice. J Parasitol. 2005; 91:1113-1115.

The following texts were included: 

Introduction

Lines 75-78. “In an experimental study, one of eight L. infantum-free male BALB/c mice used to breed females infected with this parasite tested positive for Leishmania DNA by PCR, suggesting that venereal transmission from female to male may occur in mice [19].”

Discussion

Lines 475-478. “…and the infection of an L. infantum-free male mouse used experimentally to breed females infected with this parasite support the possibility of venereal transmission from female to male [16, 19].”

Line 90: Consider substitute “… conducted between August 2015 and December 2016” using “from … to …”;

Response

Line 96. The text “This was descriptive study conducted between August 2015 and December 2016 using…” was replaced with “This was a descriptive study conducted from August 2015 to December 2016 that used…”.

Lines 93-95: What was your intention when you classified dogs among different age ranges? You did not use this anytime in your analyses. However, estrous cycle stage (i.e. estrogenic – proestrus/estrus or non-estrogenic – diestrus/anestrus) may be interesting, as showed by Magro et al. (2017). You can do this using evaluation of the status of the vaginal epithelia, and also with the ovaries (if you have). If you find different estrous cycle stages, you can make comparisons between phases. If not, forget. This may enrich your article;

Response

Our intention in classifying dogs among different age ranges was only to give a more detailed information about the characteristics of the population of dogs included in the study. As 98% of males and 100% of females were positive for L. infantum in the genital tract, the comparisons of the frequency of positivity in the genital tract between age ranges were not done.

Thanks for the suggestions of including comparisons between phases of the estrous cycle. However, we don’t have data on the estrous cycle stage of the females included in our study. Unfortunately, we did not evaluate the status of the vaginal epithelia, and we did not collect the ovaries.

Lines 97-100: See initial commentaries. Also, include a reference for the validation of the serological tests used;

Response

According to your commentaries, we have rewritten the text and included the reference Ribeiro et al. (2019) [24] for validation of serological tests used. In addition, we included the references Schubach et al. (2014) [22] and Arruda et al. (2016) [23] that describe the methodology of TR DPP® and ELISA (Biomanguinhos), respectively, and the reference Brasil (2011) that describes the criteria of Brazilian Ministry of Health [25].

The text was revised (lines 104-109) and four new references were included as already shown in response of previous commentaries.

Line 106: That normative reference is too old (2006), 15 years ago. Please, see if there is a newest one;

Response

Lines 55 and 115. The reference Brasil (2006) was updated to Brasil (2016) [1], as shown below:

References (lines 563-565)

“1. Brasil. Ministério da Saúde. Secretaria de Vigilância em Saúde. Departamento de Vigilância das Doenças Transmissíveis. Manual de vigilância, prevenção e controle de zoonoses: normas técnicas e operacionais. Brasília: Ministério da Saúde; 2016.”

Line 113: Hypotrophy is better;

Response

Methods. Line 122. The word “atrophy” was replaced with “hypotrophy”.

Lines 113-115: This classification was used before? If yes, put reference;

Response

This classification was not used before. Therefore, we did not include a reference.

Lines 119-120: Please, include reference;

Response

The following reference was included:

Line 130. Reference [28]

28-Conselho Federal de Medicina Veterinária. Guia brasileiro de boas práticas em eutanásia em animais- conceitos e procedimentos recomendados. 2013. Available: http://portal.cfmv.gov.br/uploads/files/Guia%20de%20Boas%20Pr%C3%A1ticas%20para%20Eutanasia.pdf.pdf. Accessed 25 June 2020.

Line 130: Maybe Drosophila needs to be in italics. I don’t know. Confirm;

Response

You are right. Line 144. The word “Drosophila” was written in italics.

Line 136: Exclude “collected”, and change tissues to organs. You evaluated organs, not tissues. This happens in several parts of the text. Change;

Response

The words “tissue” or “tissues” was replaced with “organ” or “organs” in all the text:

Line 147-149: I think this information needs to be referenced;

Response

Line 167. The verb “is” was replaced with “was”. 

Line 169. The information was referenced. The reference Oliveira et al. (2017) [32] was included.

Reference (lines 654-657)

32-Oliveira VDC, Boechat VC, Mendes Junior AAV, Madeira MF, Ferreira LC, Figueiredo FB, et al. Occurrence of Leishmania infantum in the central nervous system of naturally infected dogs: Parasite load, viability, co-infections and histological alterations. PLoS One. 2017;12:e0175588.

Line 156: For histopathology you put the tick of the specimen (line 164, good!). Put here too;

Response

Lines 182-183. Immunohistochemistry. The text “For IHC, the slides were submitted to …” was replaced with “Histological sections (5 μm thick) were mounted on silanized microscope slides and submitted to...”

Line 197. Histopathology. The text “…were affixed to microscopic slides…” was replaced with “…were mounted on microscope slides…”.

Lines 156-161: Did you use positive and negative controls? If yes, explain here. If not, it was not good!;

Response

Yes, we use positive and negative controls for IHC. The following text was included:

Lines 188-191. Immunohistochemistry. “Histological sections of organs intensely parasitized with amastigote forms of Leishmania were incubated with non-immune homologous serum as negative control and with polyclonal rabbit anti-Leishmania serum as positive control.”

Lines 169-171: This classification was used before? If yes, put reference;

Response

Lines 213-214. This classification was not used before. Therefore, we did not include any reference.

Line 174: “… variables were reported …;

Response

Line 219-220. The text “…variables are reported…” was replaced with “…variables were reported”. 

Line 176: Change “… CVL), a positive result in the diagnostic …” to “… CVL), positive results in the diagnostic …”;

Response

Line 221. The text “…a positive result in the diagnostic assays …” was replaced with “…positive results with diagnostic techniques…”

Line 178: Exclude “examined”;

Line 224. The word “examined” was deleted.

Line 179: Spearman’s correlation test is used for non-parametric data. You said this later, but here you need to explain why you used Spearman test. The text needs to be reordered;

Response

The text was reordered and rewritten, as indicated below:

Lines 219-233. “Data were analyzed using the free R software, version 3.5.1 [33]. The following variables were reported as simple frequencies: clinical classification (no or few clinical signs of CVL; multiple clinical signs of CVL), positive results with the diagnostic techniques, and histological alterations. Boxplots were used for comparison of log parasite loads in the samples of each organ. The Shapiro-Wilk test rejected the normality assumption of the parasite load data. Thus, the correlation between L. infantum loads in the genital tract samples was evaluated using the nonparametric Spearman correlation coefficient. The correlations ranged from -1 to 1, with positive values indicating a positive correlation and negative values an inverse correlation. In addition, the nonparametric Mann-Whitney test was applied to compare log parasite loads among all organs of the genital tract (male and female ones) and according to the intensity of clinical signs and inflammatory infiltrate. The level of agreement between the results of the diagnostic techniques (positive/negative) for detection of Leishmania infection in all organs of the genital tract was assessed by calculating Cohen’s kappa (k) statistic, considering the classification proposed by Landis and Koch [34]. A level of significance of 5% was adopted in this study.

Line 181: I don’t know if non-normal distribution is a currently used terminology. Please, confirm that;

Response

You are right. Therefore, we replaced the text “The Shapiro-Wilk test revealed a non-normal distribution of the parasite load data.” with the text “The Shapiro-Wilk test rejected the normality assumption of the parasite load data.” (lines 223-224).

Line 182-184: Considering that, at the end of the paragraph you will put the significance, P < 0.05 here is unnecessary. Also, “significance of 95%” is better;

Response

The text “(P < 0.05)” was excluded.

We prefer the term “significance of 5%” (line 233) and not the term “confidence of 95%”, because all the results of P value were expressed at this level of significance in the present study.

Line 182: Here is not clear if you compared all organs (male and female ones) among than, or if you compared male organs among than and female organs among than;

Response

Mann-Whitney test was applied to compare log parasite loads among all organs of the genital tract (male and female ones) and not to compare male organs among than and female organs among than.

The text was rewritten in order to make it clear, as shown below:

Lines 227-228. “Mann-Whitney test was applied to compare log parasite loads among all organs of the genital tract (male and female ones)…”

Lines 190-193: Exclude this phrase. It is too specific to be here, and you said this before (lines 116-120);

Response

The following phrase in the ethics statement was deleted, as recommended:

“The dogs were sedated by intramuscular administration of ketamine hydrochloride and acepromazine maleate and euthanized with an intravenous overdose of sodium thiopental and potassium chloride. All efforts were made to minimize suffering.”

Lines 196-206: I would like to see the results for males and females separately here. Something like: “Clinical examination of the 70 dogs revealed clinical signs in 59 (84.3%), x males (x% of the males) and x females (x% of the females): thinness (x males and y females), ...”;

Response

We rewrote the results, as recommended:

Lines 242-248. “Clinical examination of the 70 dogs revealed clinical signs in 59 (84.3%), 38 (54.3%) males and 21 (30.0%) females. The following clinical signs were observed: thinness (26 males and 10 females), skin ulcer (25 males and 9 females), hair loss (23 males and 10 females), onychogryphosis (20 males and 11 females), lymphadenomegaly (21 males and 10 females), furfuraceous desquamation (21 males and 5 females), splenomegaly (11 males and 7 females), cachexia (8 males and 4 females), keratoconjunctivitis (8 males and 3 females), and hepatomegaly (2 males and 2 females).”

Lines 203-204: Here I am not sure if that animal was really infected at the time of euthanasia. I think that to exclude this animal from analyses would be better;

Response

We decided to maintain the animal the dog in the study to follow the inclusion criteria, as explained above, despite it was not possible to confirm the L. infantum infection in this dog. Although this dog had no impact in the calculation of L. infantum load in the genital tract, it was important for the calculation of frequency of L. infantum infection in the genital tract of male dogs seropositive for this parasite according to the criteria of the Brazilian Ministry of Health, as explained previously.

Line 207: Figure 1: I think that a table for these results can be better than the figure 1, with organs in lines and tests in columns. However, if figure was maintained, put organs in a logical order, testis before epididymis. Also, change testes to testis in the figure;

Response

Lines 251 and 254-263. The Figure 1 was replaced with Table 1, as recommended. Therefore, all the figures and tables were renumbered.

Lines 213-219: Testis and epididymis are double organs. What did you made with this? At least one organ positive was considered “testis” positive? The same for epididymis. It should be explained in material and methods. Also, differences among right and left organ was evaluated? This can be explored aiming to establish relationships between ipsilateral organs. This is applicable for all diagnostic procedures you used;

Response

In the case of the double organs, one fragment of right and left testis and epididymis were collected and examined separately for HP e IHC. For parasitological culture and qPCR, a pool of samples containing one fragment of the right and left testis and a pool of samples containing one fragment of the right and left epididymis were examined. Therefore, for evaluation of the frequency of positivity for L. infantum by culture and qPCR and of parasite load by qPCR, the differences between right and left organ were not evaluated. However, the results by culture and qPCR was representative of the right and left organs.

For IHC and HP, the testis and epididymis were considered positive for Leishmania if amastigote forms of this parasite were detected in at least one fragment of the right or left organ. In addition, the frequencies of positivity for L. infantum in the right and left testis and epididymis were evaluated. For calculation of the frequency of each histological alteration in the testis and epididymis, the histological alteration was considered present if detected in at least one fragment of the right or left organ. For HP, the intensity of inflammatory infiltration was classified in the organ (right or left) with the most intense inflammatory infiltrate. 

In the testis of 17 dogs positive for amastigote forms of Leishmania by HP or IHC, bilateral infection was observed in 11 (65%) dogs and unilateral in four (23%). In two dogs (12%), only one testis was examined because one dog was monorchid and the other a unilateral cryptorchid. In the epididymis of 21 dogs positive for amastigote forms of Leishmania by HP or IHC, bilateral infection was observed in 14 (67%) dogs and unilateral in five (24%). In the same two dogs (9%) that were monorchid or cryptorchid, only one epididymis was examined. A bilateral inflammatory infiltrate associated with amastigote forms of Leishmania detected by HP or IHC was observed in the testis of 11 dogs and in the epididymis of 14 dogs and a unilateral infiltrate in the testis of four dogs and in the epididymis of five dogs.

In order to clarify what we did with the double organs, the following texts were included in the manuscript:

Methods.

Sample collection. Lines 136-140. “In the case of double organs, one fragment of the right and left testis and epididymis each was collected and examined separately by HP and IHC. For parasitological culture and qPCR, a pool of samples containing one fragment of the right and left testis and a pool of samples containing one fragment of the right and left epididymis were examined.”

Immunohistochemistry. Lines 191-194. “The testis and epididymis were considered positive for Leishmania if amastigote forms of this parasite were detected in at least one fragment of the right or left organ. In addition, the frequencies of positivity for amastigote forms of L. infantum in the right and left testis and epididymis were evaluated.”

Histopathology. Lines 209-216. “The testis and epididymis were considered positive for Leishmania if amastigote forms of this parasite were detected in at least one fragment of the right or left organ. For calculation of the frequency of each histological alteration in the testis and epididymis, the histological alteration was considered present if detected in at least one fragment of the right or left organ. The intensity of inflammatory infiltration was classified in the organ (right or left) with the most intense inflammatory infiltrate. In addition, the frequencies of positivity for amastigote forms of L. infantum and of inflammatory infiltrates associated with amastigote forms of this parasite in the right and left testis and epididymis were evaluated.

Results

Legend of Table 1. Lines 259-262. 

“aA pool of samples containing one fragment of the right and left testis and a pool of samples containing one fragment of the right and left epididymis were examined.”

“bThe testis and epididymis were considered positive for Leishmania if amastigote forms of this parasite were detected in at least one fragment of the right or left organ.”

Lines 277-283. “In the testis of 17 dogs positive for amastigote forms of Leishmania by HP or IHC, bilateral infection was observed in 11 (65%) dogs and unilateral in four (23%). In two dogs (12%), only one testis was examined because one dog was monorchid and the other a unilateral cryptorchid. In the epididymis of 21 dogs positive for amastigote forms of Leishmania by HP or IHC, bilateral infection was observed in 14 (67%) dogs and unilateral in five (24%). In the same two dogs (9%) that were monorchid or cryptorchid, only one epididymis was examined.”

Legend of Fig 1. Lines 303-305. “…genome equivalents/nanogram DNA (gEq/ng) in testis (a pool of right and left organs), epididymis (a pool of right and left organs), prostate, vulva, vagina, and uterus of 68 dogs positive for L. infantum DNA in the genital tract by qPCR.…”

Legend of Table 4. Line 361.

“bThe histological alteration was considered present if detected in at least one fragment of the right or left organ.”

Lines 363-365. “A bilateral inflammatory infiltrate associated with amastigote forms of Leishmania detected by HP or IHC was observed in the testis of 11 dogs and in the epididymis of 14 dogs and a unilateral infiltrate in the testis of four dogs and in the epididymis of five dogs.”

Legend of Table 5. Lines 378-379.

“cThe intensity of inflammatory infiltration was classified in the organ (right or left) with the most intense inflammatory infiltrate.”

Line 216: Change testes to testis. Make a conference in all the text;

Response

The words “testes” and “epididymides” were replaced with “testis” and epididymis”, respectively, in all the text.

Lines 217-219: You may use Kappa index to compare different tests at the same organ. Consider this possibility to enrich your results;

The kappa index was calculated, as recommended. The modifications in the text were described in a previous response to the commentaries. 

Line 224: Clarify this in material and methods;

Response

The results described in the lines xxxx were rewritten, as indicated below:

Lines 291-293. “The qPCR technique detected L. infantum DNA in at least one genital tract sample in 98% of males (n = 44) and 96% of females (n = 24). In the female negative by qPCR, vulva and vagina tested positive for amastigote forms of Leishmania by IHC.”

The following text was deleted: “Four samples, including one vulva sample, two vagina samples and one prostate sample, were negative for L. infantum DNA. However, these four samples tested positive for Leishmania by the parasitological techniques. In these cases, the species L. infantum was identified by MLEE in the culture isolates.” 

The following text was included in the methods of parasitological culture:

Lines 145-149. “The parasitological culture was considered positive if promastigote forms of Leishmania grew in the culture medium up to 30 days and were visualized by conventional optical microscopy [30]. The detailed protocol of parasite isolation in culture is registered at https://dx.doi.org/10.17504/protocols.io.22tggen.”

Lines 226-230: I am not yet convinced if comparisons among female and male organs are relevant. Why to compare epididymis and vulva, for example. Does it make sense? Comparisons among organs of the same gender make sense for me. I am not convinced but, if I am wrong, you may let these comparisons in the article;

Response

The comparisons of frequency of positivity of L. infantum and load of this parasite in the genital organs among female and male from the same endemic area are relevant and an original approach in relation to the other studies that investigate L. infantum in the genital tract of dogs. The reason is that these comparisons allows to infer if the female genital organs are parasitized as male genital organs and thus if both sexes have potential for venereal transmission, which is already proven from male to female. According to some authors, venereal transmission of L. infantum in dogs tends to be unidirectional, from male to female, due to the exclusive tropism of L. infantum for the male genital tract. These authors suggest tropism of L. infantum for the male genital tract based on the high frequency of detection of this parasite in the male genital tract and low frequencies in the female genital tract. However, the comparisons of L. infantum parasitism in the genital tract among female and male in the present study allowed to find a similar parasite load and a high frequency of active L. infantum infection in both sexes, reinforcing the hypothesis of venereal transmission also from female to male. If we did comparisons only between genital organs of the same sex, we could not have concluded that the parasitism in the genital organs of male and female were similar.

Lines 242-243: Although table 1 show the results, I think that significant results have to be showed in the text, and the others generically classified as non-significant;

Response

The following text was included to emphasizes the significant result showed in Table 3 (Table 1 in the first version of the manuscript):

Lines 314-315. “Only parasite load in the vagina was significantly associated with the number of clinical signs.”

Lines 254-256: Here I am concerned about the possibility of lesions due to other causes but leishmaniasis;

Response

Our results indicate that the histological alterations were associated with active L. infantum infection as explained previously in the item 1 of major commentaries. Even in the 19 dogs in which histological alterations were not associated with the detection of L. infantum by parasitological methods, the inflammatory reaction was probably associated with this parasite. The reason is that DNA of L. infantum was detected in the affected genital organs of these dogs. In addition, in these 19 dogs, the inflammation may have been caused indirectly by L. infantum via immunocomplex deposition [47]. 

However, following your comments, we have reviewed the results and discussion, as shown above in the response to the commentaries of the item 1. The alterations in the text were done in the lines Lines 329-330 (Results) and lines 518-525 (Discussion). 

Lines 295-296: The same observation for lines 242-243;

Response

Lines 368-370. The following text was included to emphasizes the significant results showed in Table 5: “Only in the testis and epididymis was the inflammatory infiltrate significantly associated with the parasite load in these tissues.”

Lines 365-370: From witch part of the vagina specimens were collected? Vagina is very long! I think that this explanation is too much simple. The uterus is also near to the cranial vagina, and far from caudal vagina. It depends on the region of the vagina was sampled. It was included in your experimental protocol where vagina was sampled? I think this discussion should be excluded from the manuscript;

Response

The tissue samples of vagina were collected from the caudal vagina adjacent to the vulva. This information was included in Methods, line 132 and in the discussion, line 458. 

As we specified the anatomical region of the vulva where the samples were collected, we did not exclude the discussion about the positive correlation between parasite load in in the vulva and vagina.

Lines 381-382: Considering an ascendant infection coming from penis, the next affected organ (from those exanimated) is prostate, second epididymis and, finally, testis. So, your argumentation did not support this hypothesis. Think on bacterial prostatitis in dogs. Most of them are from ascendant infection, without epididymitis and orchitis;

Response

You are right. Therefore, we reviewed our hypothesis in the discussion, as shown below:

Discussion. Lines 480-483. “In males, the infection would then spread via the hematogenous or lymphatic route from the penis, which shows a high frequency of this parasite in the prepuce, glans penis and smegma [10, 11, 13, 15], to the testis, epididymis and prostate.”

Line 384: … prepuce, penile glans and smegma. For smegma use the reference Silva L.C. et al. Detection of Leishmania infantum in the smegma of infected dogs. Arq Bras Med Vet Zootec, 66, 731-736, 2014. doi: 10.1590/1678-41626610;

Response

We rewrote the phrase and included the smegma and the reference of Silva et al. (2014) [10], as shown below:

Discussion. Lines 481-482. “…, which shows a high frequency of this parasite in the prepuce, glans penis and smegma [10, 11, 13, 15],…”

Line 385: put uterine tubes before ovaries;

Response

Line 484. We put uterine tubes before ovaries

Lines 381-387: Your discussion is contradictory. For males the infection comes from the penis to testis/epididymis “jumping” prostate, but for females infection from vulva ascend “step by step”. It seems a “convenient” explanation to your findings, but weak. This hypothesis has to be rethought. The predilection of Leishmania for one or other organ probably is explained by other reasons, maybe the availability of parasites for venereal transmission, vulva/vagina to infect dogs, and penis/prepuce and semen (coming from testis/epididymis) to infect bitches. This is only an speculation, thinking on the predilection of the parasite to skin for vector contamination;

Response

We reviewed our hypothesis in the discussion, as recommended. Thanks for the suggestions for improving this discussion. However, we don’t believe that the predilection of Leishmania for some organs of genital tract such as epididymis, testis, vulva and vagina is related to the availability of parasites for venereal or vertical transmission. Following this hypothesis, the prostate and uterus would not be the organs less parasitized in our study. The infection of the prostate would be important for contamination of semen and venereal transmission and the infection of uterus would be important for vertical transmission. I our opinion, the lower frequency and load of L. infantum in the prostate compared to the epididymis and testis observed in this study and in others [4, 11, 15] may be due to a more efficient innate and adaptive immunity against L. infantum in this organ. In females, the infection would spread via the hematogenous or lymphatic route from the vulva and vagina to the uterus, uterine tubes and ovaries. This hypothesis may explain the lower frequency and parasite load in the uterus observed in the present study and in the uterus, uterine tubes and ovaries in the study of Boechat et al. [11] when compared to the external genital organs. However, we included in the discussion that future experimental studies on the kinetics of genital tract infection with L. infantum in dogs are necessary to confirm all of the hypotheses raised in the present study.

The reviewed discussion is shown below:

Lines 460-466. “The lower frequency and load of L. infantum in the prostate compared to the epididymis and testis observed in this study and in others [4, 11, 15] may be due to a more efficient innate and adaptive immunity against L. infantum in this organ. An important mechanism of innate immunity that protects the prostate from infections is the blood-prostate barrier [42, 45]. A temperature lower than the body temperature and reduced immune responsiveness in the testis, as well as the lack of a developed mucosal immune system in the epididymis [45, 46], may predispose these organs to the infection with L. infantum.”

Lines 480-491. “In males, the infection would then spread via the hematogenous or lymphatic route from the penis, which shows a high frequency of this parasite in the prepuce, glans penis and smegma [10, 11, 13, 15], to the testis, epididymis and prostate. In females, the infection would spread via the hematogenous or lymphatic route from the vulva and vagina to the uterus, uterine tubes and ovaries. This hypothesis may explain the lower frequency and parasite load in the uterus observed in the present study and in the uterus, uterine tubes and ovaries in the study of Boechat et al. [11] when compared to the external genital organs. According to a previously reported hypothesis, venereal transmission of L. infantum from male to female dogs would occur by transfer of amastigote forms present in the glans penis, prepuce and smegma through contact with the vulvo/vaginal mucosae or with the oral mucosa of females by licking and sniffing of the preputial/penile region, as well as by ejaculation of infected semen into the female genital tract [2, 3, 10, 11].”

Lines 500-502. “However, future experimental studies on the kinetics of genital tract infection with L. infantum in dogs are necessary to confirm all of the hypotheses raised in the present study. ”

Lines 389-391: Despite these authors (2, 3 and 9) said this, it is not trough. The canine copula is not traumatic. The erection of the bulbus glandis inside vulva did not cause trauma in the vulva or vagina. It’s anecdotal. Unfortunately, I think that the authors used this argument trying to explain the bitch infection, but it is not sustainable by the mechanism of copula in dogs. Feline copula is traumatic, but not the canine copula. The literature about canine theriogenology did not classify dog's copula as traumatic. However, the friction of the contaminated surface of penis against female mucosae may be the way of Leishmania entrance inside female body. Silva et al. (2014) hypothesized that smegma may be the mainly source of Leishmania for female contamination, although not proved. This discussion and hypothesis also has to be rethought;

Response

Following your recommendations, we rewrote the hypothesis in the discussion, as indicated below: 

Discussion. Lines 487-491. “According to a previously reported hypothesis, venereal transmission of L. infantum from male to female dogs would occur by transfer of amastigote forms present in the glans penis, prepuce and smegma through contact with the vulvo/vaginal mucosae or with the oral mucosa of females by licking and sniffing of the preputial/penile region, as well as by ejaculation of infected semen into the female genital tract [2, 3, 10, 11].”

We excluded the text: “… through frequent traumatic wounds in the external genitalia that occur during copulation in dogs”.

Lines 396-399: As discussed before, I think it is plausible. Also, in the experiment where venereal transmission was showed (Silva et al., 2009), the infected dog and susceptive bitch were put together to copulate, and authors conclude that semen was the vehicle of Leishmania to the bitch. The contact of dog and bitch other than semen and male/female genitalia contact could be the source of infection, including penile-prepuce-smegma and vulva-vagina, as far as due to oral infection with contaminated fluids, if possible (not reported yet);

Response

The discussion was rewritten, as shown in the previous response to commentaries (Lines 487-491).

Lines 406-408: I think that this speculative suggestion should be accompanied by something like this: "However, it needs to be confirmed …". Please, include;

Response

Discussion. Lines 511-512. We have included the following phrase “However, this hypothesis needs to be confirmed.”

Lines 422-424: Assis et al. (2010) found poor semen characteristics in dogs with CVL, but attributed the low semen quality probably to epididymal affection. You and others found epididymitis associated to Leishmania presence in the organ. I think this could be discussed here, besides of the testis discussion, as epididymis affection seems to be more important than testis affection lowering semen quality (Assis V.P. et al. Dogs with Leishmania chagasi infection have semen abnormalities that partially revert during 150 days of Allopurinol and Amphotericin B therapy. Anim. Reprod. Sci., v.117, p.183-186, 2010. doi: 10.1016/j.anireprosci.2009.03.003). Labat et al. (2010) also showed semen abnormalities in dogs with CVL, however, they did not explain clearly how the dogs were diagnosed for CVL (Labat E. et al. Qualidade espermática de sêmen de cães naturalmente infectados por Leishmania sp. Arq Bras Med Vet Zootec, v.62, p.609-614, 2010. doi: 10.1590/S0102-09352010000300016). Consider discussing this;

Response

We reviewed the discussion according to the recommendations and included the references of Assis et al. (2010) [50] and Labat et al. (2010) [51], as shown below:

Discussion. Lines 533-539. “These testicular alterations, together with epididymitis, which were mostly bilateral, as well as prostatitis accompanied by fibrosis and glandular atrophy associated with L. infantum infection in this study, can compromise semen quality and can cause infertility of dogs with CVL [9, 50, 51]. According to Assis et al. [50], poor semen quality of L. chagasi (syn. L. infantum)-infected dogs, which is characterized by poor motility, principal piece defects and detached heads, indicates epididymal dysfunction.”

Lines 437-438: It cannot be placed in conclusion, but you can put at the end of the discussion.

Response

We removed the conclusion “The castration of dogs would therefore be an important control measure of CVL in endemic areas.” and included this text in a new phrase of the discussion in the lines 444-446, as shown below:

“Considering the high frequency of active L. infantum infection in the genital tract of male and female dogs in this study, the castration of dogs would be an important control measure of CVL in endemic areas, preventing venereal and vertical transmission.”

Reviewer #3: The manuscript provides sounding results concerning the presence of Leishmania parasite in the genital tract of male and female dogs. The manuscript is relevant with supporting results. Thus, it should be accepted for publication.

6. PLOS authors have the option to publish the peer review history of their article (what does this mean?). If published, this will include your full peer review and any attached files.

Do you want your identity to be public for this peer review? For information about this choice, including consent withdrawal, please see our Privacy Policy.

Reviewer #1: Yes: Angamuthu Selvapandiyan

Reviewer #2: No

Reviewer #3: Yes: Claudio Vieira da SilvaWhile revising your submission, please upload your figure files to the Preflight Analysis and Conversion Engine (PACE) digital diagnostic tool, https://pacev2.apexcovantage.com/. PACE helps ensure that figures meet PLOS requirements. To use PACE, you must first register as a user. Registration is free. Then, login and navigate to the UPLOAD tab, where you will find detailed instructions on how to use the tool. If you encounter any issues or have any questions when using PACE, please email PLOS at figures@plos.org. Please note that Supporting Information files do not need this step.

---

## [Decision Letter · Decision Letter 1]

12 Aug 2020

Frequency, active infection and load of Leishmania infantum and associated histological alterations in the genital tract of male and female dogs

PONE-D-20-10750R1

Dear Dr. Menezes,

We’re pleased to inform you that your manuscript has been judged scientifically suitable for publication and will be formally accepted for publication once it meets all outstanding technical requirements.

Kind regards,

Farhat Afrin, Ph.D.

Academic Editor

PLOS ONE

Additional Editor Comments (optional):

Reviewers' comments:

Reviewer's Responses to Questions

**Comments to the Author**

1. If the authors have adequately addressed your comments raised in a previous round of review and you feel that this manuscript is now acceptable for publication, you may indicate that here to bypass the “Comments to the Author” section, enter your conflict of interest statement in the “Confidential to Editor” section, and submit your "Accept" recommendation.

Reviewer #2: (No Response)

2. Is the manuscript technically sound, and do the data support the conclusions?

Reviewer #2: (No Response)

3. Has the statistical analysis been performed appropriately and rigorously? 

Reviewer #2: (No Response)

4. Have the authors made all data underlying the findings in their manuscript fully available?

Reviewer #2: (No Response)

5. Is the manuscript presented in an intelligible fashion and written in standard English?

Reviewer #2: (No Response)

6. Review Comments to the Author

Reviewer #2: (No Response)

7. PLOS authors have the option to publish the peer review history of their article (what does this mean?). If published, this will include your full peer review and any attached files.

Reviewer #2: **Yes: **Guilherme Ribeiro Valle

---

## [Editor Report · Acceptance letter]

14 Aug 2020

PONE-D-20-10750R1 

Frequency, active infection and load of Leishmania infantum and associated histological alterations in the genital tract of male and female dogs 

Dear Dr. Menezes:

I'm pleased to inform you that your manuscript has been deemed suitable for publication in PLOS ONE. Congratulations! Your manuscript is now with our production department. 

Kind regards, 

on behalf of

Dr. Farhat Afrin 

Academic Editor

PLOS ONE